# Foliar Applications of Salicylic Acid on Boosting Salt Stress Tolerance in Sour Passion Fruit in Two Cropping Cycles

**DOI:** 10.3390/plants12102023

**Published:** 2023-05-18

**Authors:** Thiago Galvão Sobrinho, André Alisson Rodrigues da Silva, Geovani Soares de Lima, Vera Lúcia Antunes de Lima, Vitória Ediclécia Borges, Kheila Gomes Nunes, Lauriane Almeida dos Anjos Soares, Luciano Marcelo Fallé Saboya, Hans Raj Gheyi, Josivanda Palmeira Gomes, Pedro Dantas Fernandes, Carlos Alberto Vieira de Azevedo

**Affiliations:** 1Post Graduate Program Agricultural Engineering, Federal University of Campina Grande, Campina Grande 58430-380, PB, Brazil; thiago.galvao@estudante.ufcg.edu.br (T.G.S.); andre.alisson@estudante.ufcg.edu.br (A.A.R.d.S.); vera.lucia@professor.ufcg.edu.br (V.L.A.d.L.); vitoria.ediclecia@estudante.ufcg.edu.br (V.E.B.); kheila.gomes@estudante.ufcg.edu.br (K.G.N.); luciano.marcelo@professor.ufcg.edu.br (L.M.F.S.); hans@pq.cnpq.br (H.R.G.); pedro.dantas@professor.ufcg.edu.br (P.D.F.); carlos.azevedo@professor.ufcg.edu.br (C.A.V.d.A.); 2Academic Unit of Agrarian Sciences, Federal University of Campina Grande, Pombal 58840-000, PB, Brazil; lauriane.almeida@professor.edu.br

**Keywords:** *Passiflora edulis* Sims, brackish water, fitormônio

## Abstract

Brazil stands out as the largest producer of sour passion fruit; however, the water available for irrigation is mostly saline, which can limit its cultivation. This study was carried out with the objective of evaluating the effects of salicylic acid in the induction of tolerance in sour passion fruit to salt stress. The assay was conducted in a protected environment, using a completely randomized design in a split-plot scheme, with the levels of electrical conductivity of the irrigation water (0.8, 1.6, 2.4, 3.2, and 4.0 dS m^−1^) considering the plots and concentrations of salicylic acid (0, 1.2, 2.4, and 3.6 mM) the subplots, with three replications. The physiological indices, production components, and postharvest quality of sour passion fruit were negatively affected by the increase in the electrical conductivity of irrigation water, and the effects of salt stress were intensified in the second cycle. In the first cycle, the foliar application of salicylic acid at concentrations between 1.0 and 1.4 mM partially reduced the harmful effects of salt stress on the relative water content of leaves, electrolyte leakage, gas exchange, and synthesis of photosynthetic pigments, in addition to promoting an increase in the yield and quality parameters of sour passion fruit.

## 1. Introduction

In recent years, the availability of fresh water has caused concerns, especially in semi-arid regions [1]. Climate change that has occurred all over the world has increased atmospheric temperatures and caused droughts [2,3]. Thus, the climatic imbalance between evaporation and precipitation rates promotes the increase in salt concentrations in sources of water used for irrigation [4,5].

High salinity in water and/or soil causes salt stress, one of the environmental stresses that most compromise sustainability and agricultural production worldwide [6]. Salt stress causes damage to agricultural production and inhibits crop growth due to reduced water availability for plants, caused by the decrease in the osmotic potential of the soil solution, leading to stomatal closure and compromising transpiration and the CO_2_ assimilation rate [7]. Salinity reduces chlorophyll synthesis and photochemical efficiency, limiting the process of photosynthesis, which directly affects crop production [8]. According to Dias et al. [9], irrigation with saline water also reduces pulp yield and compromises the postharvest quality of fruits, as observed by Andrade et al. [10], who found reductions in the number of fruits and in the average weight of sour passion fruit irrigated by water with electrical conductivity (ECw) above 0.7 dS m^−1^, while Lima et al. [11] found reductions in the polar and equatorial diameter of passion fruit as a function of salt stress; in another study, Ramos et al. [12] observed reductions in pulp yield and postharvest quality of passion fruit irrigated with ECw above 0.6 dS m^−1^.

Recent studies have shown that irrigation with saline water can compromise the production components of passion fruit and reduce its yield [13,14,15]. However, the severity of the effect of salt stress on plants depends on other factors, such as genotype, time of exposure to stress, edaphoclimatic conditions of the region, irrigation management, fertilization, and foliar application of elicitor substances [3,16].

Thus, studies that enable the use of saline waters in irrigated agriculture are important to ensure the sustainability of crops. In this context, the use of salicylic acid has emerged as a promising alternative for minimizing the deleterious effects caused by salt stress on plants [17,18,19].

Salicylic acid is a natural phenolic compound, which acts as a non-enzymatic antioxidant and an endogenous signaling molecule, inducing tolerance to salt stress [20]. Its beneficial effects are dependent on the concentration, plant species, stage of crop development, and method of application [3,21,22].

Several studies have reported that foliar spraying with salicylic acid can attenuate the deleterious effects caused by salt stress in strawberry [23], American grapevine [24], date palm [25], orange [26], and soursop [18]. However, information on its use in sour passion fruit crops irrigated with saline water is incipient in the literature.

Sour passion fruit (*Passiflora edulis* Sims) is the most recognized and cultivated *Passifloraceae* species in Brazil, especially in the Northeast region [27]. Brazil stands out as the largest producer and consumer of passion fruit in the world, producing 683,993 tons in 2021 in an area of 44,827 ha, achieving an average yield of 15.26 t ha^−1^, with the Northeast region accounting for 69.6% (476,006 tons) of the national production [28].

Due to the physical-chemical quality and acceptance by consumers, its fruits are used for fresh consumption and/or through agro-industrial processing in the preparation of carbonated and mixed drinks, syrups, jellies, dairy products, ice cream and canned foods [29,30]. It should be noted that passion fruit is considered a salt-sensitive crop [31], with an irrigation water salinity threshold of 1.3 dS m^−1^ [32,33].

This study is based on the hypothesis that foliar application of salicylic acid at adequate concentrations mitigates the deleterious effects caused by irrigation with saline waters on the physiology, production components, and postharvest quality of the fruits of sour passion fruit, inducing the tolerance of plants to salt stress by increasing the biosynthesis of organic compounds and modulating activities of enzymes that detoxify reactive oxygen species. In view of the above, the aim of this study was to evaluate the effect of foliar application of salicylic acid concentrations on the induction of tolerance of sour passion fruit to salt stress in two cropping cycles.

## 2. Results 

### 2.1. First Cycle

The relative water content and electrolyte leakage in the leaf blade were significantly influenced (*p* ≤ 0.01) by the interaction between ECw × SA (Table 1).

Higher ECw reduced the relative water content in leaves (Figure 1A). However, foliar application of salicylic acid with concentrations up to 1.2 mM promoted an increase in the relative water content. The highest value of RWC (90.02%) was obtained in plants irrigated with an ECw of 0.8 dS m^−1^ and under a salicylic acid concentration of 1.2 mM, corresponding to an increase of 1.15% compared to plants grown with the same salinity level (0.8 dS m^−1^) without the application of salicylic acid (0 mM). On the other hand, foliar application of SA at concentrations greater than 1.2 mM compounded the deleterious effects of salt stress on the relative water content, with the lowest value (68.07%) obtained in plants irrigated with an ECw of 4.0 dS m^−1^ and under a salicylic acid concentration of 3.6 mM.

The electrolyte leakage in the leaf blade was increased as the ECw rose above 1.4 dS m^−1^ (Figure 1B), regardless of salicylic acid concentration. Application of salicylic acid at concentrations greater than 1.0 mM intensified the deleterious effects of salt stress, and the highest value of % EL (38.19%) was obtained in plants irrigated with an ECw of 4.0 dS m^−1^ and under a SA concentration of 3.6 mM. However, sour passion fruit plants subjected to the highest level of ECw (4.0 dS m^−1^) and a SA concentration of 1.0 mM showed % EL of 34.37%, that is, reduction of 10.0% compared to plants grown with the same ECw and application of SA at concentration of 3.6 mM, demonstrating the beneficial effect of salicylic acid in the acclimatization to salt stress, when applied at appropriate concentrations.

The interaction between ECw × SA significantly (*p* ≤ 0.01) influenced leaf gas exchange (Table 2) at 180 DAT.

The internal CO_2_ concentration (Figure 2A) was reduced by the application of salicylic acid up to a concentration of 1.2 mM, regardless of the ECw level. The lowest value of internal CO_2_ concentration (120.89 μmol mol^−1^) was recorded in plants irrigated with an ECw of 0.8 dS m^−1^ and under a salicylic acid concentration of 1.2 mM. Sour passion fruit plants irrigated with an ECw of 0.8 dS m^−1^ and under a SA concentration of 1.2 mM showed a reduction of 5.06% (6.44 μmol mol^−1^) in the internal CO_2_ concentration compared to plants grown with the same ECw without the application of SA (0 mM). The highest internal CO_2_ concentration (316.31 μmol mol^−1^) was recorded in plants irrigated with an ECw of 4.0 dS m^−1^ and a salicylic acid concentration of 3.6 mM.

Foliar spraying of salicylic acid with concentrations up to 1.2 mM promoted an increase in *gs* (Figure 2B), even when plants were subjected to the highest level of ECw (4.0 dS m^−1^). The highest value of *gs* (0.246 mol H_2_O m^−2^ s^−1^) was obtained in plants cultivated with an ECw of 1.5 dS m^−1^ at a SA concentration of 1.2 mM, corresponding to an increase of 4.24% (0.01 mol H_2_O m^−2^ s^−1^) compared to plants grown with the same salinity level (1.5 dS m^−1^) and without the application of salicylic acid (0 mM). However, foliar spraying of SA at concentrations greater than 1.2 mM intensified the deleterious effects of salt stress on *gs*, and the lowest value (0.152 mol H_2_O m^−2^ s^−1^) was obtained in plants subjected to an ECw of 4.0 dS m^−1^ and a SA concentration of 3.6 mM.

The transpiration and CO_2_ assimilation rate of sour passion fruit were also favored by foliar spraying of salicylic acid up to a concentration of 1.2 mM, regardless of the electrical conductivity of irrigation water (Figure 2C,D). Plants subjected to a concentration of 1.2 mM and irrigated with an ECw of 2.0 dS m^−1^ obtained the highest values of transpiration (2.39 mmol H_2_O m^−2^ s^−1^) and CO_2_ assimilation rate (5.74 μmol CO_2_ m^−2^ s^−1^). When comparing in relative terms the transpiration and CO_2_ assimilation rate of plants irrigated with an ECw of 2.0 dS m^−1^ and subjected to a salicylic acid concentration of 1.2 mM with the values of plants cultivated with the same salinity level without application of SA (0 mM), increments of 4.82% (0.11 mmol H_2_O m^−2^ s^−1^) and 4.36% (0.24 μmol CO_2_ m^−2^ s^−1^) were observed, respectively.

There was a significant effect (*p* ≤ 0.05) of the interaction between the ECw × SA for chlorophyll *a*, chlorophyll *b*, and total chlorophyll (Table 3). On the other hand, the levels of electrical conductivity of irrigation water significantly (*p* ≤ 0.01) influenced all the variables of photosynthetic pigments, while the concentrations of salicylic acid as a single factor affected all variables, except the carotenoid contents of sour passion fruit at 180 DAT.

Foliar application of SA with concentrations up to 1.0 mM promoted increments in the synthesis of chlorophyll *a*, chlorophyll *b*, and total chlorophyll, regardless of the ECw (Figure 3A–C). Plants subjected to a SA concentration of 1.0 mM and irrigated with water of 1.4 dS m^−1^ obtained the highest values of Chl *a* (740.97 μg mL^−1^), Chl *b* (220.34 μg mL^−1^), and Chl *t* (952.41 μg mL^−1^). When comparing in relative terms the contents of chlorophyll *a*, *b* and total chlorophyll of plants irrigated with water of 1.4 dS m^−1^ and subjected to SA concentration of 1.0 mM with those of plants cultivated with the same level of salinity without application of SA (0 mM), increments of 3.38% (24.23 μg mL^−1^), 5.09% (10.67 μg mL^−1^), and 3.71% (34.09 μg mL^−1^) were observed, respectively. However, foliar spraying of salicylic acid at concentrations greater than 1.0 mM intensified the deleterious effects of salt stress, and the lowest contents of Chl *a* (400.57 μg mL^−1^), Chl *b* (108.28 μg mL^−1^), and Chl *t* (508.82 μg mL^−1^) were obtained in plants irrigated with an ECw of 4.0 dS m^−1^ and under a SA concentration of 3.6 mM.

The biosynthesis of carotenoids (Figure 3D) of sour passion fruit was negatively affected by the increase in the electrical conductivity of irrigation water, with reductions of 10.17% per unit increment of ECw. When comparing the carotenoid content of plants irrigated with an ECw of 4.0 dS m^−1^ to that of plants subjected to an ECw of 0.8 dS m^−1^, a reduction of 35.44% (80.92 μg mL^−1^) was observed.

The interaction between the ECw × SA did not significantly (*p* > 0.05) influence the chlorophyll fluorescence variables (Table 4). The levels of electrical conductivity of irrigation water significantly (*p* ≤ 0.01) affected the initial fluorescence, maximum fluorescence, variable fluorescence, and quantum efficiency of photosystem II of sour passion fruit. Salicylic acid concentrations, in turn, did not cause significant effects on any of the chlorophyll fluorescence variables.

The increase in the ECw caused an increasing linear effect on the initial fluorescence of sour passion fruits (Figure 4A), with an increment of 2.31% per unit increase of ECw. Plants irrigated with an ECw of 4.0 dS m^−1^ had an increase of 7.27% (46.9), compared to those cultivated with an ECw of 0.8 dS m^−1^. Unlike the effect observed in the initial fluorescence (Figure 4A), the maximum fluorescence was reduced with increasing ECw (Figure 4B). Plants cultivated with an ECw of 4.0 dS m^−1^ had a reduction of 10.88% (250.93), compared to those cultivated with an ECw of 0.8 dS m^−1^.

The variable fluorescence (Figure 4C) of sour passion fruit plants decreased with the increase in the electrical conductivity of irrigation water. Plants irrigated with an ECw of 0.8 dS m^−1^ had a variable fluorescence of 1782.03, while the lowest value (1471.76) was obtained under an ECw of 4.0 dS m^−1^, that is, there was a reduction of 310.27 (17.41%) under the highest salinity level (4.0 dS m^−1^). A similar effect was observed in the quantum efficiency of photosystem II (Figure 4D), that is, a reduction in Fv/Fm with the increase in the electrical conductivity of irrigation water, with the lowest value of Fv/Fm (0.716) in plants irrigated with an ECw of 4.0 dS m^−1^, corresponding to a reduction of 7.25% (0.056) compared to plants irrigated with an ECw of 0.8 dS m^−1^.

The interaction between the ECw × SA did not significantly influence the production components of sour passion fruits in the first cycle (Table 5). On the other hand, the levels of electrical conductivity of irrigation water significantly (*p* ≤ 0.01) influenced all the variables of the production components, while the concentrations of salicylic acid affected the number of fruits per plant, average fruit weight, and total production per plant.

The increase in ECw negatively affected the number of fruits per plant of sour passion fruit (Figure 5A), with reductions of 15.91% per unit increment of ECw, with the lowest value of NFP (10.86 fruits per plant) obtained in plants irrigated with an ECw of 4.0 dS m^−1^, whereas the highest value of NFP (26.08 fruits per plant) was obtained in those subjected to an ECw of 0.8 dS m^−1^. NFP was also influenced by the concentrations of salicylic acid (Figure 5B); foliar application of salicylic acid at the estimated concentration of 1.4 mM led to the highest value of NFP (19.84 fruits per plant), corresponding to an increase of 12.48% (2.2 fruits per plant) compared to plants subjected to a SA concentration of 0 mM. The average fruit weight (Figure 5C) and total production per plant (Figure 5E) were also reduced by the increase in the electrical conductivity of irrigation water, with reductions of 9.41% in average fruit weight and 18.95% in total production per plant per unit of ECw increase. When comparing the AFW and TPP of plants irrigated with an ECw of 4.0 dS m^−1^ to the values of those subjected to an ECw of 0.8 dS m^−1^, reductions of 32.55% (92.71 g per fruit) and 71.46% (4867.2 g per plant) were observed in the average fruit weight and total production per plant, respectively.

Foliar spraying of salicylic acid at the estimated concentration of 1.48 mM promoted the highest value of AFW (257.80 g per fruit), corresponding to an increase of 4.33% (10.69 g per fruit) compared to the control treatment (0 mM) (Figure 5D). The estimated SA concentration of 1.35 mM stood out in relation to the total production per plant, leading to the value of 6209.7 g per plant, that is, an increase of 6.13% (358.5 g per plant) compared to plants subjected to a SA concentration of 0 mM (Figure 5F).

The irrigation water salinity negatively affected the polar and equatorial diameter of the fruits of sour passion fruit (Figure 6A,B), with reductions of 7.86% in the polar diameter and 3.66% in the equatorial diameter per unit of ECw increment. When comparing the FPD and FED of plants irrigated with ECw of 4.0 dS m^−1^ to those of plants subjected to ECw of 0.8 dS m^−1^, reductions of 26.84% (25.74 mm) and 12.05% (8.40 mm) were observed in polar and equatorial diameter, respectively.

There was a significant effect (Table 6) in the interaction between the levels of electrical conductivity of irrigation water and salicylic acid concentrations on the hydrogen potential and ascorbic acid contents of sour passion fruits. The levels of electrical conductivity of irrigation water and salicylic acid concentrations significantly (*p* ≤ 0.01) affected the hydrogen potential, soluble solids, ascorbic acid, and titratable acidity of sour passion fruits in the first cycle.

The increase in the ECw reduced the pH of the pulp of sour passion fruits (Figure 7A), regardless of the concentration of salicylic acid, with the lowest pH value (2.47) recorded in plants irrigated with an ECw of 4.0 dS m^−1^ and under a SA concentration of 3.6 mM. However, it was observed that the foliar spraying of salicylic acid up to a concentration of 1.6 mM promoted an increase in pH, with the highest pH value (3.52) obtained in plants subjected to an ECw of 0.8 dS m^−1^ and a SA concentration of 1.6 mM. The foliar spraying of salicylic acid at a concentration of 1.6 mM also promoted an increase in the ascorbic acid content (Figure 7B) of sour passion fruits, with the highest value of ascorbic acid (4.44 mg 100 g^−1^ pulp) observed in plants irrigated with an ECw of 1.9 dS m^−1^, while plants irrigated with the same level of ECw (1.9 dS m^−1^) and subjected to a SA concentration of 0 mM showed a reduction of 4.73% (0.21 mg 100 g^−1^ pulp) in ascorbic acid content in the pulp, compared to those subjected to a SA concentration of 1.6 mM.

The increase in the ECw caused a positive linear effect on the soluble solids content in the pulp of sour passion fruits (Figure 7C), with an increase of 6.43% per unit of ECw increase. Plants irrigated with an ECw of 4.0 dS m^−1^ had an increase of 19.86% (2.58 °Brix) compared to those cultivated with an ECw of 0.8 dS m^−1^. Soluble solids were also influenced by the concentrations of salicylic acid (Figure 7D); plants subjected to an estimated SA concentration of 1.5 mM stood out with the highest value of soluble solids (14.98 °Brix), corresponding to an increase of 5.27% (0.66 °Brix) compared to plants subjected to a SA concentration of 0 mM. 

The titratable acidity of sour passion fruit pulp increased with the increase in the electrical conductivity of irrigation water (Figure 8A). There was an increase of 8.37% per unit increment of ECw. The highest titratable acidity (6.98%) was recorded in plants irrigated with an ECw of 4.0 dS m^−1^, while the lowest value (5.58%) was obtained in plants subjected to irrigation with an ECw of 0.8 dS m^−1^. Application of salicylic acid up to the estimated concentration of 1.64 mM promoted an increase in the titratable acidity (Figure 8B), with reduction in titratable acidity at SA concentrations greater than 1.64 mM. The highest value obtained (7.12%) was recorded in plants subjected to an estimated concentration of 1.64 mM, whereas plants that did not receive an application of salicylic acid (0 mM) had a titratable acidity of 6.29%.

### 2.2. Second Cycle

Relative water content and electrolyte leakage in the leaf blade were significantly (*p* ≤ 0.01) influenced only by the levels of electrical conductivity of irrigation water (Table 7), that is, the SA concentrations as a single factor and its interaction with ECw × SA did not influence (*p* > 0.05) RWC and % EL at 360 DAT.

The increase in the ECw caused a reduction in the relative water content in the leaves of sour passion fruit (Figure 9A), with a decrease of 6.21% per unit increment of ECw; when comparing plants irrigated with an ECw of 4.0 dS m^−1^ with those irrigated with an ECw of 0.8 dS m^−1^, a reduction of 20.91% was observed. The electrolyte leakage in the leaf blade was also negatively affected by the increase in the ECw (Figure 9B), and the highest value of % EL (37.72%) was obtained in plants irrigated with an ECw of 4.0 dS m^−1^.

The interaction between the ECw × SA did not significantly influence (*p* > 0.05) the gas exchange at 360 DAT (Table 8). There was a significant effect (*p* ≤ 0.05) only of the levels of electrical conductivity of irrigation water on the internal CO_2_ concentration, stomatal conductance, transpiration, and CO_2_ assimilation rate.

The increase in the electrical conductivity of irrigation water promoted an increase in the internal CO_2_ concentration of sour passion fruit (Figure 10A), equal to 69.42% per unit increment of ECw. When comparing in relative terms the *Ci* of plants irrigated with water of the highest ECw (4.0 dS m^−1^) to that of plants subjected to the lowest salinity level (0.8 dS m^−1^), an increase of 142.82% (154.68 μmol mol^−1^) was observed. On the other hand, stomatal conductance was reduced linearly with the increase in the level of electrical conductivity of irrigation water (Figure 10B), with a decrease of 10.18% per unit increment of ECw. When comparing the *gs* of plants subjected to the highest level of ECw (4.0 dS m^−1^) to that of plants cultivated under an ECw of 0.8 dS m^−1^, a decrease of 35.47% (0.073 mol H_2_O m^−2^ s^−1^) was observed. Salt stress, caused by the increase in electrical conductivity of irrigation water, negatively affected transpiration (Figure 10C) and CO_2_ assimilation rate (Figure 10D), with reductions of 8.82% in transpiration and 8.52% in CO_2_ assimilation rate, per unit of ECw increase. When comparing the *E* and *A* of plants irrigated with an ECw of 4.0 dS m^−1^ to the values of those subjected to an ECw of 0.8 dS m^−1^, reductions of 30.36% (0.624 mol H_2_O m^−2^ s^−1^) and 29.27% (1.37 μmol CO_2_ m^−2^ s^−1^) were observed in transpiration and CO_2_ assimilation rate, respectively.

Thee hundred and sixty days after transplanting, the contents of chlorophyll *a*, chlorophyll *b*, total chlorophyll, and carotenoids of sour passion fruit were significantly (*p* ≤ 0.01) affected only by the levels of electrical conductivity of irrigation water (Table 9).

The synthesis of chlorophyll *a* (Figure 11A), chlorophyll *b* (Figure 11B), total chlorophyll (Figure 11C), and carotenoids (Figure 11D) was reduced by the increase in the electrical conductivity of irrigation water, by 8.39% (Chl *a*), 10.94% (Chl *b*), 8.04% (Chl *t*), and 10.04% (Car) per unit increment of ECw. Plants irrigated with ECw of 4.0 dS m^−1^ showed reductions of 28.76% (147.09 μg mL^−1^), 38.37% (55.22 μg mL^−1^), 27.49% (176.41 μg mL^−1^), and 34.94% (68.39 μg mL^−1^) in Chl *a*, Chl *b*, Chl *t*, and Car, respectively, when compared to plants subjected to the lowest salinity level (0.8 dS m^−1^).

As observed in the first cycle (Table 10), the initial fluorescence, maximum fluorescence, variable fluorescence, and quantum efficiency of photosystem II of sour passion fruits were significantly (*p* ≤ 0.01) affected only by the levels of ECw at 360 DAT.

The increase in the ECw promoted an increasing linear effect on the initial fluorescence of sour passion fruit plants (Figure 12A), with an increase of 2.27% per unit of ECw increase. Plants grown with an ECw of 4.0 dS m^−1^ had an increase of 7.13% (49.9) compared to those cultivated with an ECw of 0.8 dS m^−1^. Unlike the effect observed in the initial fluorescence (Figure 12A), the maximum fluorescence was reduced with the increase in the electrical conductivity of irrigation water (Figure 12B). Plants irrigated with an ECw of 4.0 dS m^−1^ had a reduction of 13.31% (178.0) compared to those subjected to an ECw of 0.8 dS m^−1^. The variable fluorescence (Figure 12C) of sour passion fruit also decreased with the increase in the electrical conductivity of irrigation water. Plants irrigated with an ECw of 0.8 dS m^−1^ had a variable fluorescence of 1030.76, while the lowest value (779.78) was obtained under an ECw of 4.0 dS m^−1^; that is, there was a reduction of 250.98 (24.35%) under the highest level of ECw (4.0 dS m^−1^).

The quantum efficiency of photosystem II (Figure 12D) decreased with the increase in the electrical conductivity of irrigation water, with the lowest value of Fv/Fm (0.671) obtained in plants irrigated with an ECw of 4.0 dS m^−1^, corresponding to a reduction of 12.86% (0.099) compared to those irrigated with an ECw of 0.8 dS m^−1^.

The interaction between the ECw and the concentrations of SA did not significantly influence the production components of sour passion fruits in the second cycle (Table 11). The levels of electrical conductivity of irrigation water significantly (*p* ≤ 0.01) influenced all variables, while the concentrations of salicylic acid affected the number of fruits per plant, average fruit weight, and total production per plant.

As observed in the first cycle, the increase in the electrical conductivity of irrigation water negatively affected the number of fruits per plant of sour passion fruit (Figure 13A), with a reduction of 15.69% per unit increment of ECw, with the lowest NFP value (7.76 fruits per plant) recorded in plants irrigated with an ECw of 4.0 dS m^−1^, corresponding to a reduction of 28.57% (3099 fruits per plant) compared to the first cycle, demonstrating that the effects of salt stress are intensified with greater exposure to stress, reflecting the accumulation of salts in the soil. The average fruit weight (Figure 13B) and total production per plant (Figure 13C) were also reduced by the increase in the electrical conductivity of irrigation water, with reductions of 11.02% in the average fruit weight and 19.67% in total production per plant per unit of ECw increase. When comparing the AFW and TPP of plants irrigated with an ECw of 4.0 dS m^−1^ to the values of those subjected to an ECw of 0.8 dS m^−1^, reductions of 38.66% (104.74 g per fruit) and 71.46% (4867.2 g per plant) were observed in the average fruit weight and total production per plant, respectively. The salt stress caused by the increase in the electrical conductivity of irrigation water also reduced the polar diameter (Figure 13D) and equatorial diameter (Figure 13E) of fruits of sour passion fruit in the second cycle, with reductions of 7.83% in the fruits’ polar diameter and 4.86% in the fruits’ equatorial diameter per unit of ECw increase. When comparing the FPD and FED of plants irrigated with an ECw of 4.0 dS m^−1^ to the values of those subjected to an ECw of 0.8 dS m^−1^, reductions of 26.73% (27.68 mm) and 17.31% (12.80 mm) were observed in the polar and equatorial diameter, respectively.

According to the summary of the analysis of variance (Table 12) in the second production cycle, the variables hydrogen potential, ascorbic acid, soluble solids, and titratable acidity of sour passion fruit were significantly (*p* ≤ 0.01) affected only by the levels of ECw.

The pH (Figure 14A) and ascorbic acid content (Figure 14B) in the pulp of sour passion fruit were reduced by the increase in the ECw, with a decrease of 8.99% (pH) and 10.19% (AA) per unit increment of ECw. Plants irrigated with an ECw of 4.0 dS m^−1^ showed reductions of 31.09% (1.2) in pH and 35.43% (1.52 mg 100 g^−1^ pulp) in the ascorbic acid content. Unlike the effect observed on pH and AA (Figure 14A,B), the soluble solids (Figure 14C) and titratable acidity (Figure 14D) increased with the increase in the electrical conductivity of irrigation water by 7.03% (SS) and 19.65% (TA) per unit increment of ECw. Plants irrigated with an ECw of 4.0 dS m^−1^ had increments of 21.37% (2.70 °Brix) in soluble solids and 54.29% in titratable acidity compared to those cultivated with an ECw of 0.8 dS m^−1^.

## 3. Discussion

Salt stress caused by excess salts in water and/or soil limits plant growth and development [34] and negatively affects ionic homeostasis and, consequently, nutrient absorption, causing losses in agricultural production worldwide, especially in arid and semi-arid regions [35,36,37]. Fruit crops generally have a greater sensitivity to salt stress than annual crops [38].

The results obtained in this study show that the salt stress caused by the increase in the electrical conductivity of irrigation water negatively affected the leaf water status, observed by the relative water content, promoted an increase in the electrolyte leakage in the leaf blade, and reduced gas exchange, synthesis of photosynthetic pigments and chlorophyll *a* fluorescence, negatively affecting the production components and postharvest quality of sour passion fruits.

Two studies have indicated that the sensitivity and tolerance of crops to salt stress may vary between species and cultivars of the same species, in addition to depending on climatic conditions, type of soil, plant development stage, irrigation method, and time of exposure to salt stress [16,39]. In the present study, it was observed that the deleterious effects of salinity were intensified in the second cycle, possibly due to a longer period of exposure to salt stress and the accumulation of salts in the soil.

In the first cycle of sour passion fruit, foliar application of salicylic acid between the estimated concentrations of 1.0 and 1.4 mM was able to partially mitigate the effects of salt stress on the relative water content in the leaf blade, electrolyte leakage, gas exchange, and photosynthetic pigments. In addition, salicylic acid promoted an increase in the production components and postharvest quality of the fruits of sour passion fruit. In the second cycle, there was no effect of the application of salicylic acid on any of the variables analyzed, which may be related to the lower number of applications due to the reduction in the cycle, as the first cycle lasted 258 days, while the second cycle lasted 172 days, that is, a reduction of 86 days.

The decrease in the relative water content in the leaves (Figure 1A and Figure 9A) and the increase in electrolyte leakage (Figure 1B and Figure 9B) with the increase in ECw can be seen as a harmful effect of salinity. The reduction in RWC results from the loss of turgor of plant tissues, since salinity causes osmotic stress, which hinders the absorption and translocation of water from the soil to the plant, affecting its growth and metabolism [40]. Salt stress also favors the production of reactive oxygen species (ROS), such as superoxide radical, hydroxyl radicals, and hydrogen peroxide [41]. Under normal conditions, the production and elimination of ROS are balanced, but under salt stress conditions, there is an imbalance between production and elimination, which can cause photo-oxidative damage to photosystems and the cell membrane peroxidation [42,43], generating an increase in the percentage of electrolyte leakage, as observed in the present study (Figure 1B and Figure 9B).

In a study conducted by Silva Neta et al. [44] with the sour passion fruit ‘BRS Rubi do Cerrado’ under irrigation with saline water (ECw ranging from 0.3 to 3.5 dS m^−1^), a reduction in RWC was also observed with the increase in the levels of electrical conductivity of irrigation water, equal to 7.73% when comparing plants irrigated with ECw of 3.5 dS m^−1^ to those cultivated with an ECw of 0.3 dS m^−1^. Wanderley et al. [45] evaluated yellow passion fruit under salt stress (ECw ranging from 0.3 to 3.1 dS m^−1^) and observed an increase of 24.65% in the percentage of electrolyte leakage in plants irrigated with an ECw of 3.1 dS m^−1^ compared to those cultivated with an ECw of 0.3 dS m^−1^.

Despite the deleterious effects of salt stress on RWC and EL, foliar application of salicylic acid in concentrations between 1.0 and 1.2 mM mitigated the effects of irrigation water salinity in the first cropping cycle of sour passion fruit. Salicylic acid acts to protect plant cells from the toxicity caused by ion accumulation and improves antioxidant activity, nitrogen metabolism, and water absorption in plants [46,47], increasing leaf turgor and reducing cell membrane damage [1].

The results obtained in the present study indicate that salt stress caused by the increase in the electrical conductivity of irrigation water negatively impacted the gas exchange variables (*Ci*, *gs*, *E* and *A*) of sour passion fruit in both cycles. When subjected to salt stress, plants tend to close their stomata in order to reduce water loss to the atmosphere [18]. In addition, the reduction in leaf turgor, as observed in this study (Figure 1A and Figure 9A), induces stomatal closure, as a defense mechanism of the plant against the loss of water by transpiration, affecting physiological processes, such as stomatal conductance, internal CO_2_ concentration and CO_2_ assimilation rate [48]. Reductions in gas exchange in sour passion fruit plants caused by irrigation with saline water have also been reported in other studies, such as [5,13,16,30].

Stomata are the structures responsible for regulating the gas exchange of plants [49]. In this study, the first cycle of sour passion fruit foliar application of salicylic acid at a concentration of 1.2 mM increased stomatal conductance, resulting in improvements in the internal CO_2_ concentration, transpiration, and CO_2_ assimilation rate (Figure 2). Previous studies have shown that salicylic acid can reduce lipid peroxidation and interact with other plant hormones, increasing the tolerance of plants to salt stress [50,51]. In addition, the beneficial effects of salicylic acid observed in gas exchanges may be related to the accumulation of osmoprotectants, improving the turgor of plant cells under stress and the activation of antioxidant enzymes, resulting in better photosynthetic activity [52].

Photosynthetic pigments are considered determinant elements for plant growth and development [53]. The results of this study showed that the increase in electrical conductivity of irrigation water negatively affected the photosynthetic pigments of sour passion fruit, especially in the second cycle. The excess salts present in the irrigation water inhibits the activity of 5-aminolevulinic acid, which is a precursor of chlorophyll, in addition to increasing the activity of the chlorophyllase enzyme, which acts by degrading the molecules of photosynthetic pigments, causing damage to chloroplasts and limiting the activity of pigmentation proteins [54,55]. Reductions in photosynthetic pigments in sour passion fruit plants as a function of irrigation with saline water have also been reported in studies conducted by [14,56,57].

The foliar application of salicylic acid at the estimated concentration of 1.0 mM mitigated the effects of salt stress on the synthesis of chlorophyll *a*, chlorophyll *b,* and total chlorophyll in sour passion fruits in the first cropping cycle. According to Hundare et al. [58], salicylic acid can stimulate the biosynthesis of chlorophyll and/or reduce its degradation, improving plant growth and development. In agreement with the present study, Silva et al. [34] evaluated the effect of foliar application of salicylic acid on photosynthetic pigments in soursop under salt stress and found that foliar application of salicylic acid at a concentration of 1.4 mM reduced the effects of irrigation water salinity, promoting an increase in the contents of photosynthetic pigments (Chl *a*, Chl *b*, Chl *t*, and Car).

Chlorophyll fluorescence variables are widely used to explain chlorophyll energy dissipation in thylakoid membranes [59,60]. In the present study, it was observed that the concentrations of salicylic acid were not able to attenuate the effects of salt stress on chlorophyll fluorescence in any of the cropping cycles. The increase in the electrical conductivity of irrigation water promoted an increase in initial fluorescence (Figure 4 and Figure 12), indicating damage to the light-harvesting complex of the photosystem II of sour passion fruit. According to Kalaji et al. [61], the increase in initial fluorescence results in the utilization of less photochemical energy in the reaction centers of photosystem II, thus serving as an indicator of salt stress effects.

In turn, maximum fluorescence and variable fluorescence were reduced by the increase in the electrical conductivity of irrigation water. These results may be related to low efficiency in quinone photoreduction and electron flow between photosystems, which results in low PSII activity in the thylakoid membrane, directly influencing the flow of electrons between photosystems [62,63]. In addition, it may indicate that the photosynthetic apparatus was damaged by salt stress, compromising photosystem II, with negative effects on the photosynthetic process [1], as observed in this study.

The reductions in maximum (Fm) and variable (Fv) fluorescence caused by the increase in the electrical conductivity of irrigation water contributed negatively to reducing the quantum efficiency of photosystem II (Fv/Fm). Several authors consider Fv/Fm values between 0.75 and 0.85 as normal in unstressed plants [18,64,65]. Thus, the results indicate that irrigation with an ECw above 2.0 dS m^−1^ in the first cycle and above 1.4 dS m^−1^ in the second cycle negatively affected Fv/Fm as the values were lower than 0.75.

The reduction in the quantum efficiency of photosystem II of sour passion fruits shows lower activity of P_680_ under salt stress conditions, which may be associated with the activity of the chlorophyllase enzyme, which reduces the chlorophyll contents and, consequently, affects the electron capture and transport between the reaction centers to free plastoquinone [66]. Reduction in Fv/Fm in sour passion fruit plants due to salt stress was also verified by Andrade et al. [56], who found a decrease of 6.36% per unit of ECw increase, highlighting it as indicative of the occurrence of a photoinhibition effect caused by salt stress.

The increase in the electrical conductivity of irrigation water negatively affected the production components of sour passion fruits, observed by the number of fruits per plant, average fruit weight, total production per plant, and polar and equatorial diameter of fruits, with the most intense reductions in the second cycle. Under salt stress, plants suffer from water and nutrient deficiencies due to the osmotic effect that hinders the absorption of water and nutrients [67]. The excess of salts in irrigation water also induces the inhibition of physiological and metabolic processes, negatively affecting the production components [68]. Reductions in the production components of sour passion fruits as a function of irrigation water salinity have also been reported in studies conducted by [13,15,39].

Despite the reduction in the production components of sour passion fruits, it was observed in this study that the foliar application of salicylic acid at concentrations between 1.4 and 1.5 mM promoted an increase in the number of fruits per plant, average fruit weight, and total production per plant (Figure 5) in the first cropping cycle. In this study, it was observed that salicylic acid regulated stomatal conductance (Figure 2B) and increased photosynthetic activity (Figure 2B) and the synthesis of chlorophyll (Figure 3), contributing directly to an increase in the production components. The beneficial effect of salicylic acid on the production components of sour passion fruit may be related to its role in reducing the absorption of Na^+^ and increasing that of N, P, K, Ca, and Mg by plants [69].

The increase in the electrical conductivity of irrigation water reduced the pH of sour passion fruit pulp both in the first and in the second cycle. According to Lacerda et al. [19], pH is an important variable of postharvest quality, as low values can ensure the conservation of the pulp without the need for high heat treatment, thus avoiding nutritional losses. In the first cropping cycle, foliar application of salicylic acid at the estimated concentration of 1.6 mM increased the ascorbic acid contents of the sour passion fruit pulp (Figure 7B). In the second cycle, the ascorbic acid contents were reduced by irrigation water salinity from 0.8 dS m^−1^.

Soluble solids content is a parameter that has been used as an indicator of fruit quality [70]. The increase in the electrical conductivity of irrigation water increased the soluble solids content of sour passion fruits, whose values were within the ideal range from 13 to 15 °Brix, as described by Aguiar et al. [71]. Unlike the results obtained in this study, Ramos et al. [12] evaluated the production and postharvest quality of passion fruit irrigated with saline water (ECw ranging from 0.6 to 3.0 dS m^−1^) and found reductions in soluble solids contents with the increase in the electrical conductivity of irrigation water.

Titratable acidity in the pulp of sour passion fruit increased with the increase in the electrical conductivity of irrigation water. Titratable acidity is an important chemical attribute for the preservation of food products for both the consumer and the industry, as it makes the food more resistant to deterioration by microorganisms and allows greater flexibility in the addition of sugar, which is of particular importance in the preparation of ready-to-drink beverages [72,73].

In general, the results obtained indicate that irrigation with an ECw of up to 3.0 dS m^−1^ does not reduce the postharvest quality of the fruits of sour passion fruit, as they had values of pH, soluble solids, and titratable acidity within the technical norms for identity and quality standards for passion fruit pulp issued by the Brazilian Ministry of Agriculture, which establishes pH between 2.70 and 3.80, soluble solids greater than 11 °Brix, and titratable acidity above 2.50%.

Based on the data of production per plant obtained in the first and second cycle, the tolerance of sour passion fruit to irrigation water salinity was determined through the relative production, obtained by the Plateau model followed by linear decay (Figure 15), obtaining a salinity threshold of 0.8 dS m^−1^, with a reduction of 14.78% per unit increment of ECw above this level. It is possible to obtain a relative production of 70% with an ECw of 2.83 dS m^−1^ and a relative production of 50% with an ECw of 4.18 dS m^−1^. The use of an electrical conductivity of irrigation water equal to or greater than 7.57 dS m^−1^ will result in a relative production of 0%.

According to the criteria of degrees of tolerance based on relative production, cited by Fageria et al. [75], considering the percentage of production loss obtained at an ECw of 4.0 dS m^−1^ compared to plants irrigated with an ECw of 0.8 dS m^−1^, sour passion fruit is classified as sensitive to irrigation water salinity, with a reduction of 72.81%.

In general, it was found in this study that irrigation with an ECw above 0.8 dS m^−1^ reduced the relative water content in the leaf blade and increased electrolyte leakage, a fact that directly reflected on gas exchange and on the biosynthesis of photosynthetic pigments; chlorophyll fluorescence was also negatively impacted, which compromised production and affected the postharvest quality of the passion fruit, especially in the second cycle. However, it was verified that the use of saline water in the cultivation of passion fruit in a semi-arid region can be made possible by the foliar application of salicylic acid in adequate concentrations. Furthermore, it was observed that SA concentrations greater than 1.4 mM can intensify the deleterious effects of saline stress on plants under semi-arid conditions. 

## 4. Materials and Methods

### 4.1. Location of the Experiment

The study was conducted in a protected environment belonging to the Agricultural Engineering Department of the Federal University of Campina Grande, in Campina Grande, Paraíba, Brazil, located at 07°15′18″ S, 35°52′28″ W, at an altitude of 550 m. Meteorological data collected during the experimental period are shown in Figure 16.

### 4.2. Treatments and Experimental Design

A completely randomized experimental design was adopted in a split-plot scheme, with the levels of electrical conductivity of the irrigation water—ECw (0.8, 1.6, 2.4, 3.2, and 4.0 dS m^−1^) considering the plots and acid concentrations salicylic acid—SA (0, 1.2, 2.4, and 3.6 mM) subplots, with three replications and one plant per plot (Figure 17). Concentrations of salicylic acid were applied by foliar spray.

The concentrations of salicylic acid used in this study were based on a study conducted with soursop [3], while the levels of electrical conductivity of irrigation water were adapted from the study conducted by [13] with the yellow passion fruit crop.

### 4.3. Experiment Setup and Conduction 

In the experiment, seeds of the sour passion fruit cultivar ‘BRS GA1’ were used. The seedlings were formed by sowing three seeds in plastic bags measuring 15 × 20 cm. The bags were filled with a substrate composed of 84% soil, 15% sand, and 1% humus (*v*/*v*). The seedlings were transplanted to the drainage lysimeters 70 days after sowing (DAS). The experiment was conducted using plastic pots adapted as drainage lysimeters, with a capacity of 200 L, filled with a 1.0 kg layer of crushed stone followed by 250 kg of soil classified as Entisol, collected at 0–30 cm depth in the municipality of Lagoa Seca, PB, whose physical-chemical characteristics were determined according to the methodologies described by Teixeira et al. [76]: Potential of hydrogen (1:2.5 soil/water) = 6.50, organic matter = 8.10 g dm^−3^, P = 79 mg dm^−3^, K^+^, Na^+^, Ca^2+^, Mg^2+^, Al^3+^ + H^+^ equivalent to 0.24, 0.51, 14.90, 5.4, and 0.90 cmol_c_ kg^−1^, Granulometric fraction: sand, silt, clay = 572.7, 100.70, 326.60 g kg^−1^, respectively, Moisture content(dag kg^−1^) at field capacity (33.42 kPa) and permanent wilting (1519.5 kPa) = 25.91, 12.96, respectively.

Irrigation waters with different levels of electrical conductivity were prepared by dissolving NaCl, CaCl_2_·2H_2_O, and MgCl_2_·6H_2_O salts, in the equivalent ratio of 7:2:1, respectively, in local-supply water (ECw = 0.38 dS m^−1^). This ratio is commonly found in sources of water used for irrigation in small farms in the Northeast [77]. The irrigation waters were prepared considering the relationship between ECw and salt concentration [78], according to Equation (1):(1)Q ≅ 10×ECw
where:Q—quantity of salts to be added (mmol_c_ L^−1^)ECw—electrical conductivity of water (dS m^−1^)

Thirty days after transplantation, irrigation with saline water began, with a 2-day irrigation interval, applying the water in each lysimeter according to treatment in order to maintain soil moisture close to field capacity, and the volume to be applied was determined according to the water needs of the plants, estimated by the water balance, as indicated by Equation (2):(2)$$\mathrm{VI}\;=\frac{\left(\mathrm{Va}\;-\;\mathrm{Vd}\right)}{\left(1\;-\;\mathrm{FL}\right)}\;$$
where:VI—volume of water to be used in the irrigation event (mL);Va—volume applied in the previous irrigation event (mL);Vd—volume drained (mL);LF—leaching fraction of 0.15, applied every 30 days to avoid excessive accumulation of salts.

Fertilizations with nitrogen, phosphorus, and potassium were based on the methodology proposed by Costa and Silva [79] for the passion fruit crop, with application of 50 g of single superphosphate per plant as basal. Nitrogen fertilization (225 g per plant) and potassium fertilization (345 g per plant) were split into 18 portions and applied at 15-day intervals via fertigation, using urea as a source of nitrogen and potassium chloride as source of potassium. Micronutrient applications were performed fortnightly using a Dripsol^®^ micro solution to meet the nutritional requirements, at a concentration of 1.0 g L^−1^, containing Mg (1.1%), Zn (4.2%), B (0.85%), Fe (3.4%), Mn (3.2%), Cu (0.5%), and Mo (0.05%), through the leaves on the adaxial and abaxial sides with a backpack sprayer.

The spacing used was 2 m between rows and 1.5 m between plants in a trellis system (2.5 m height). The plants were trained on a vertical trellis with nylon string. When the plants reached 20 cm above the trellis, the apical bud was pruned to stimulate the growth of two secondary branches, which were trained in opposite directions until reaching 0.75 m in length. After the secondary branches reached the previously established length, a new pruning of the apical bud was performed to stimulate the growth of the tertiary branches, which were trained up to 30 cm from the ground. Throughout the experiment, tendrils were removed to favor the development of the crop (Figure 18).

The second cycle began after the crown cleaning pruning (at 258 DAT) carried out at the end of the harvest of the first cycle, with the purpose of promoting aeration and entry of sunlight, as well as the renewal of productive branches, eliminating dead, old, diseased, and/or unproductive branches, reducing the problems caused by pests and diseases, improving the phytosanitary status of plants, and also facilitating cultural practices, in particular fertilization and irrigation. Tertiary and quaternary branches were pruned about 40 cm away from the wire.

Solutions of salicylic acid at adequate concentration were obtained by dissolution in 30% ethyl alcohol, prepared immediately before each application event. The first application was performed 15 days after transplanting the seedlings between 17:00 and 18:00 h; the other applications were made at intervals of 30 days, until the beginning of the flowering stage, spraying the leaves so as to completely wet the leaves, between 17:00 and 18:00 h, using a backpack sprayer (Jacto XP^®^—Jacto, Pompeia, SP, Brazil) with a capacity of 12 L, a working pressure (maximum) of 88 psi (6 bar) and a JD 12P nozzle. During the spraying of salicylic acid, a structure with plastic tarpaulin was used to prevent the solution from drifting onto neighboring plants.

### 4.4. Variables Analyzed

At 180 (first cycle) and 360 (second cycle) days after transplantation (DAT), the relative water content, electrolyte leakage in the leaf blade, gas exchange, photosynthetic pigments, and chlorophyll a fluorescence were evaluated. At the time of harvest, the following parameters were evaluated: production components—number of fruits per plant, average fruit weight, total production, polar and equatorial diameter of fruits, and postharvest variables—pH, titratable acidity, ascorbic acid content, and soluble solids.

Relative water content (RWC) was determined using the methodology described by Weatherley [80]. Electrolyte leakage (% EL) was determined using the methodology of Scotti-Campos et al. [81]. Photosynthetic pigments (chlorophyll *a*, chlorophyll *b*, total chlorophyll, and carotenoids) were quantified following the laboratory method developed by Arnon [82], expressed in μg mL^−1^.

Leaf gas exchange, represented by internal CO_2_ concentration (*Ci*, μmol CO_2_ m^−2^ s^−1^), stomatal conductance (*gs*, mol H_2_O m^−2^ s^−1^), transpiration (*E*, mmol H_2_O m^−2^ s^−1^), and CO_2_ assimilation rate (*A*, μmol CO_2_ m^−2^ s^−1^), was measured on the third leaf, counted from the apex of the main branch of the plant, using a irradiation of 1200 μmol photons m^−2^ s^−1^ and an air flow of 200 mL min^−1^, using the portable photosynthesis meter “LCPro+” from ADC BioScientific Ltd. Leaf gas exchange measurements were performed between 08:00 and 10:00 h, under conditions of ambient temperature and CO_2_ concentration.

Chlorophyll fluorescence was evaluated in the third leaf, counted from the apex of the main branch of the plant, at 08:00 h, with the pulse-modulated fluorimeter OS5p from Opti Science, using the Fv/Fm protocol to determine the following variables: initial fluorescence (F0), maximum fluorescence (Fm), variable fluorescence (Fv = Fm − F0), and quantum efficiency of photosystem II (Fv/Fm). This protocol was performed after the adaptation of the leaves to the dark for a period of 30 min, using a clip of the device, in order to ensure that all acceptors were oxidized, that is, with the reaction centers open.

Ripe fruits (with yellow peel) were harvested from 190 to 258 DAT in the first cycle and from 380 to 430 DAT in the second cycle. After harvest, the number of fruits per plant (NFP, unit per plant), average fruit weight (AFW, g per fruit), total production per plant (TPP, g per plant), fruit polar diameter (FPD, mm) and fruit equatorial diameter (FED, mm) were determined. Then, postharvest analyses of sour passion fruit fruits were performed. The pH was determined directly in the pulp immediately after harvest, with a digital meter previously calibrated at pH 7.0 with a buffer solution, soluble solids (°Brix) were measured by direct reading on a digital refractometer, and ascorbic acid content (mg per 100 g of pulp) was determined by titration. The determinations were performed using the methodologies recommended by IAL [83]. Titratable acidity was measured according to the standards of IAL [83] and expressed as a percentage of citric acid.

### 4.5. Tolerance

Through the data of total production per plant in the two cycles, the tolerance level of sour passion fruit to salt stress was determined based on the relative yield, using the plateau followed by a linear decline model proposed by Maas and Hoffman [74]. The model parameters were fitted by minimizing the square of errors with the Microsoft Excel Solver tool, as reported by Bione et al. [84]. The plants were classified according to the degree of tolerance, adopting the criterion of reduction in relative yield Fageria and Gheyi [75], with four levels of classification: T (tolerant; decrease between 0 and 20%), MT (moderately tolerant; decrease between 20% and 40%), MS (moderately sensitive; decrease between 40% and 60%), and S (Sensitive; decrease > 60%). The percentage of loss was based on the total production per plant determined under the highest ECw level (4.0 dS m^−1^) compared to the condition with the lowest ECw (0.8 dS m^−1^).

### 4.6. Statistical Analysis

The collected data were first subjected to the distribution normality test (Shapiro–Wilk test). Subsequently, analysis of variance was performed at 0.05 probability level and, in cases of significance, regression analysis. Statistica v. 7.0 software was used for statistical analyses [85]. The choice of the model was based on the coefficients of determination (R^2^). In case of significance of the interaction between factors, SigmaPlot software v.14.5 was used to create the response surfaces.

## 5. Conclusions

The physiology, production components, and postharvest quality of the fruits of sour passion fruit were negatively affected by the increase in the electrical conductivity of irrigation water above 0.8 dS m^−1^, and the effects of salt stress were intensified in the second cropping cycle. In the first cycle, the foliar application of salicylic acid at concentrations between 1.0 and 1.4 mM partially reduces the harmful effects of salt stress on the leaf relative water content, electrolyte leakage, gas exchange, and synthesis of photosynthetic pigments, in addition to promoting an increase in the number of fruits, average fruit weight, and total production per plant. Salicylic acid at concentrations greater than 1.4 mM intensifies the effects of salt stress on sour passion fruit in the first production cycle. The irrigation water salinity threshold for the cultivation of sour passion fruit is 0.8 dS m^−1^, with a reduction of 14.78% per unit increment of ECw, from this threshold. Sour passion fruit is classified as sensitive to irrigation water salinity, with a decrease of 72.81% when irrigated with an ECw of 4.0 dS m^−1^.

The results obtained in this study confirm the hypothesis that salicylic acid, when applied at appropriate concentrations, can act as a signaling molecule and reduce the effects of salt stress on sour passion fruit, with the beneficial effect depending on the concentration and crop development stage, and its application is not recommended for more than one cropping cycle. However, more studies are needed to understand how salicylic acid acts on the signaling of salt stress, in addition to validating the results in field research.

## Figures and Tables

**Figure 1 plants-12-02023-f001:**
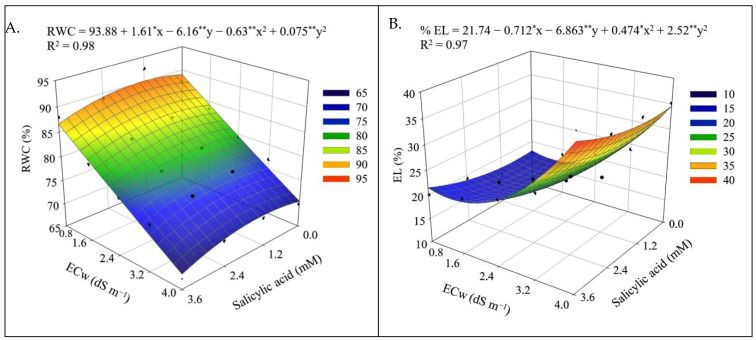
Relative water content—RWC (**A**) and electrolyte leakage in the leaf blade—EL (**B**) of sour passion fruit as a function of the interaction between the levels of electrical conductivity of irrigation water (ECw) and salicylic acid concentrations, at 180 days after transplanting. X and Y-concentration of salicylic acid and ECw, respectively, * and ** significant at *p* ≤ 0.05 and ≤0.01, respectively.

**Figure 2 plants-12-02023-f002:**
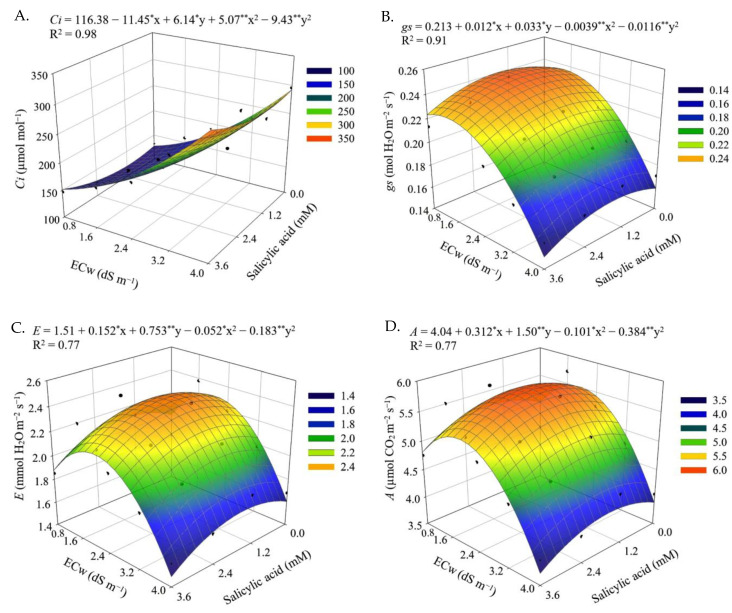
Internal CO_2_ concentration—*Ci* (**A**), stomatal conductance—*gs* (**B**), transpiration—*E* (**C**) and CO_2_ assimilation rate—*A* (**D**) of sour passion fruit as a function of the interaction between the electrical conductivity of irrigation water (ECw) and salicylic acid concentrations, at 180 days after transplanting. X and Y-concentration of salicylic acid and ECw, respectively, * and ** significant at *p* ≤ 0.05 and ≤0.01, respectively.

**Figure 3 plants-12-02023-f003:**
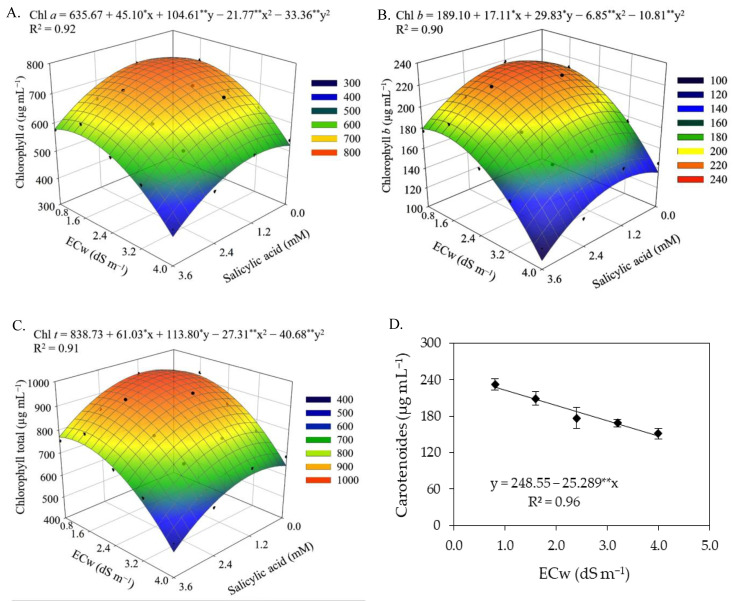
Contents of chlorophyll *a*—Chl *a* (**A**), chlorophyll *b*—Chl *b* (**B**), and total chlorophyll—Chl *t* (**C**) of sour passion fruit as a function of the ECw and SA interaction, and carotenoids (**D**) as a function of ECw levels at 180 days after transplanting. X and Y-concentration of salicylic acid and ECw, respectively, * and ** significant at *p* ≤ 0.05 and ≤0.01, respectively. Vertical lines in Figure (**D**) represent standard error of the mean (*n* = 3).

**Figure 4 plants-12-02023-f004:**
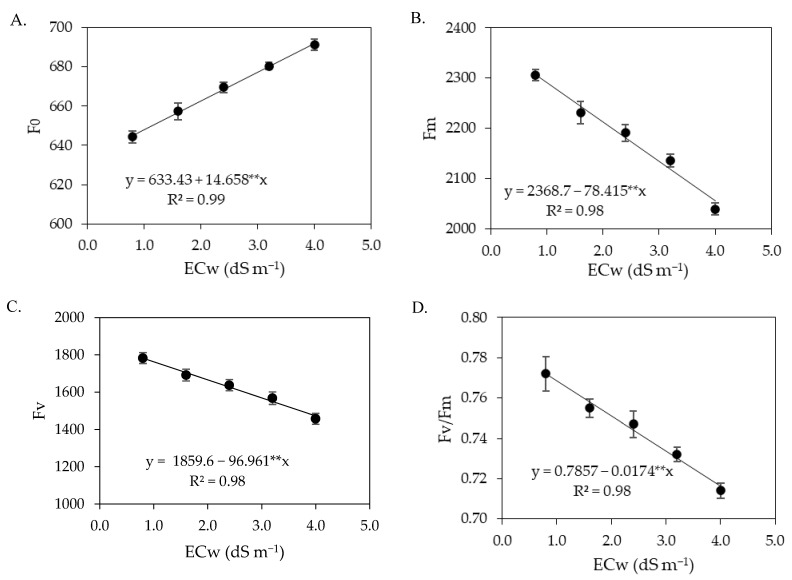
Initial fluorescence—F_0_ (**A**), maximum fluorescence—Fm (**B**), variable fluorescence—Fv (**C**), and quantum efficiency of photosystem II—Fv/Fm (**D**) of sour passion fruit as a function of the levels of electrical conductivity of irrigation water (ECw) at 180 days after transplanting. ** significant at *p* ≤ 0.01. Vertical lines represent standard error of the mean (*n* = 3).

**Figure 5 plants-12-02023-f005:**
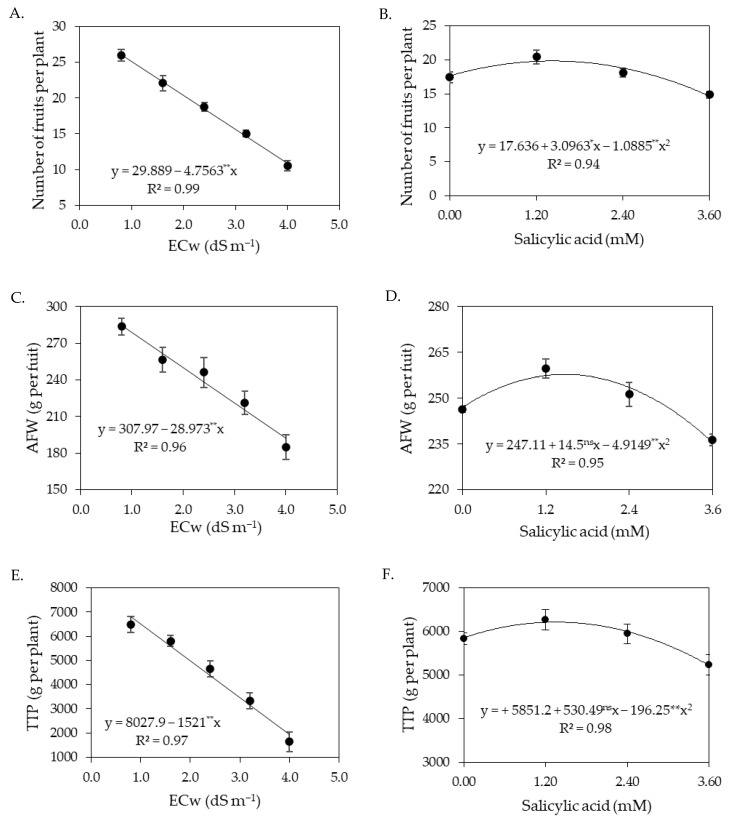
Number of fruits per plant—NFP (**A**), average fruit weight—AFW (**C**), and total production per plant—TPP (**E**) of sour passion fruit as a function of the electrical conductivity of irrigation water (ECw); and number of fruits per plant—NFP (**B**), average fruit weight—AFW (**D**), and total production per plant—TPP (**F**) as a function of salicylic acid concentrations in the first production cycle. ns, * and ** represent respectively, not significant, significant at *p* ≤ 0.05, and *p* ≤ 0.01. Vertical lines represent standard error of the mean (*n* = 3).

**Figure 6 plants-12-02023-f006:**
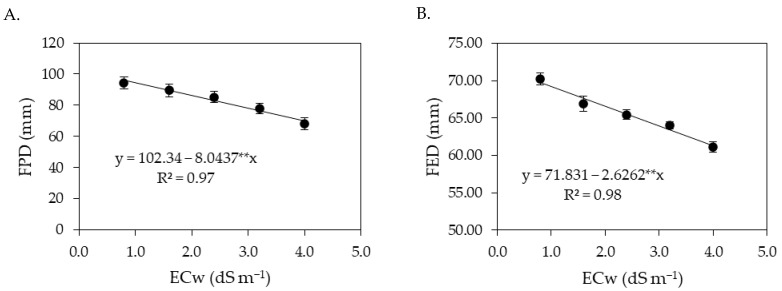
Polar (**A**) and equatorial (**B**) diameter of fruits of sour passion fruit as a function of the electrical conductivity of irrigation water (ECw). ** significant at *p* ≤ 0.01. Vertical lines represent standard error of the mean (*n* = 3).

**Figure 7 plants-12-02023-f007:**
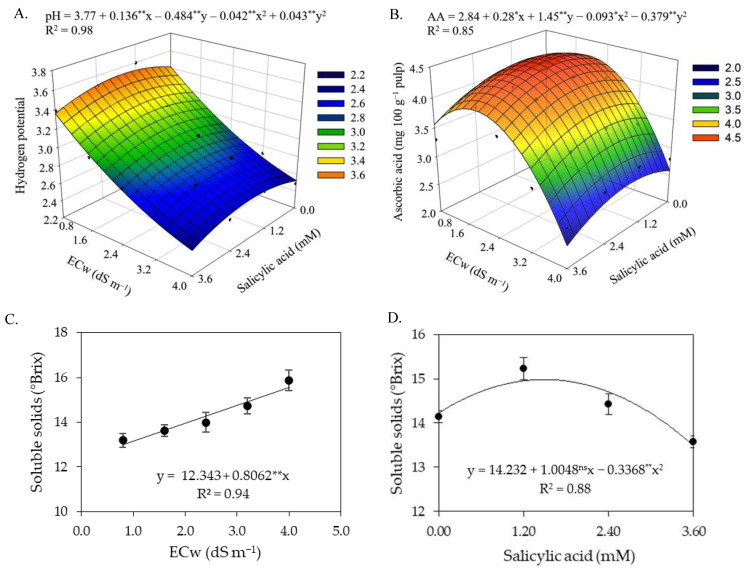
Hydrogen potential—pH (**A**) and ascorbic acid—AA (**B**) in pulp of sour passion fruit as a function of the interaction between the electrical conductivity of irrigation water (ECw) and salicylic acid (SA), and soluble solids (SS) as a function of ECw levels (**C**) and SA concentrations (**D**). X and Y-concentration of salicylic acid and ECw, respectively; ns, * and ** represent respectively, not significant, significant at *p* ≤ 0.05 and *p* ≤ 0.01. Vertical lines represent standard error of the mean (*n* = 3).

**Figure 8 plants-12-02023-f008:**
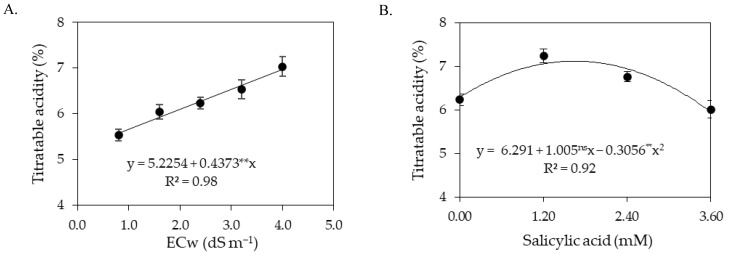
Titratable acidity in fruits of sour passion fruit as a function of the levels of electrical conductivity of irrigation water (**A**) and concentrations of salicylic acid (**B**). ns and ** respectively not significant, significant at *p* ≤ 0.01. Vertical lines represent standard error of the mean (*n* = 3).

**Figure 9 plants-12-02023-f009:**
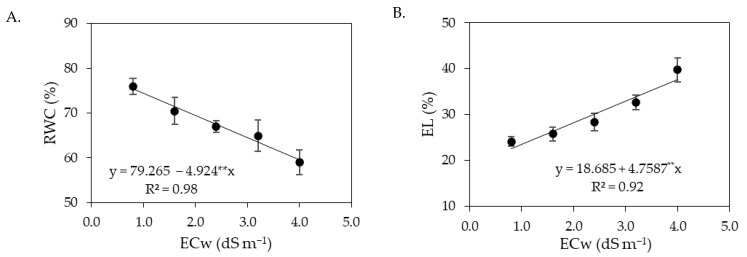
Relative water content—RWC (**A**) and electrolyte leakage in the leaf blade—% EL (**B**) of sour passion fruit as a function of the levels of electrical conductivity of irrigation water (ECw), at 360 days after transplanting. ** Significant at *p* ≤ 0.01. Vertical lines represent standard error of the mean (*n* = 3).

**Figure 10 plants-12-02023-f010:**
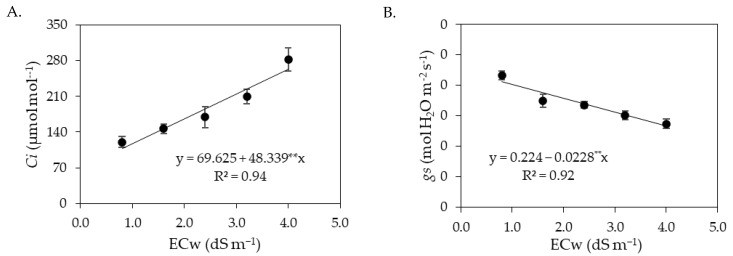
Internal CO_2_ concentration—*Ci* (**A**), stomatal conductance—*gs* (**B**), transpiration—*E* (**C**), and CO_2_ assimilation rate—*A* (**D**) of sour passion fruit as a function of the levels of electrical conductivity of irrigation water (ECw), at 360 days after transplanting. ** Significant at *p* ≤ 0.01. Vertical lines represent standard error of the mean (*n* = 3).

**Figure 11 plants-12-02023-f011:**
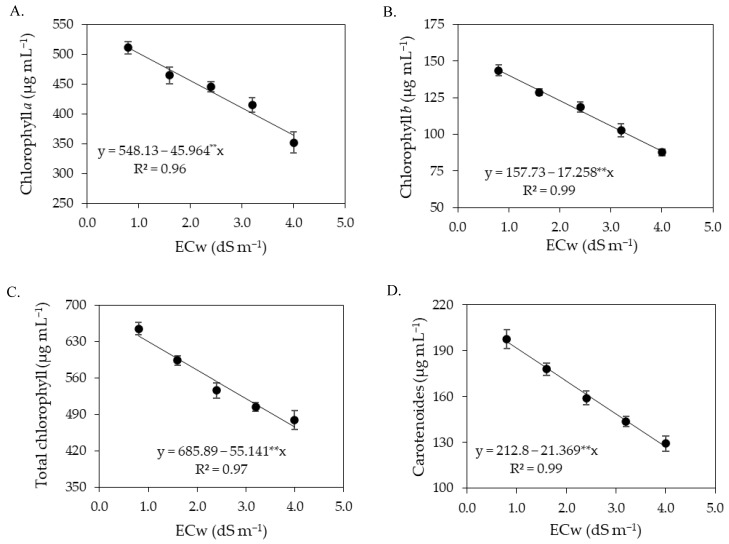
Contents of chlorophyll *a*—Chl *a* (**A**), chlorophyll *b*—Chl *b* (**B**), total chlorophyll—Chl *t* (**C**), and carotenoids (**D**) of sour passion fruit as a function of the levels of ECw, at 360 days after transplanting. ** Significant at *p* ≤ 0.01. Vertical lines represent standard error of the mean (*n* = 3).

**Figure 12 plants-12-02023-f012:**
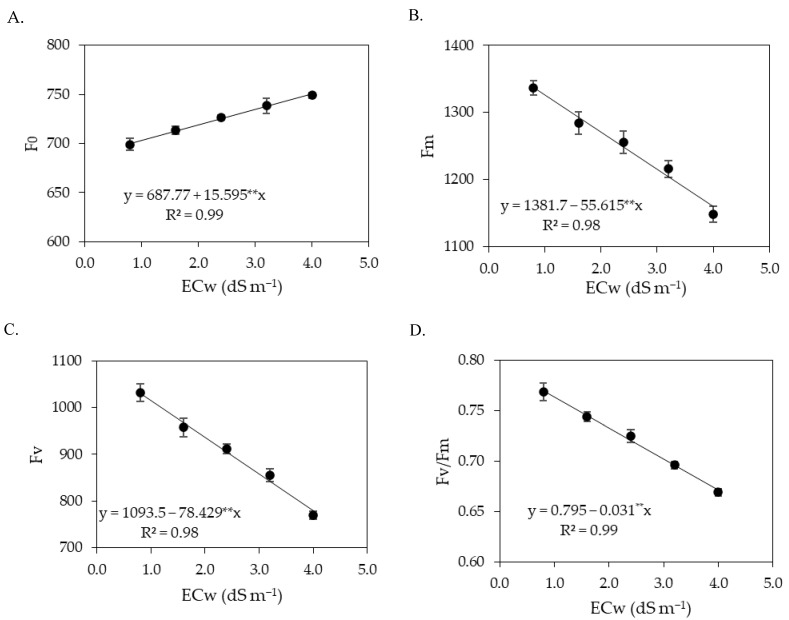
Initial fluorescence—F_0_ (**A**), maximum fluorescence—Fm (**B**), variable fluorescence—Fv (**C**), and quantum efficiency of photosystem II—Fv/Fm (**D**) of sour passion fruit as a function of the levels of ECw at 360 days after transplanting. ** Significant at *p* ≤ 0.01. Vertical lines represent standard error of the mean (*n* = 3).

**Figure 13 plants-12-02023-f013:**
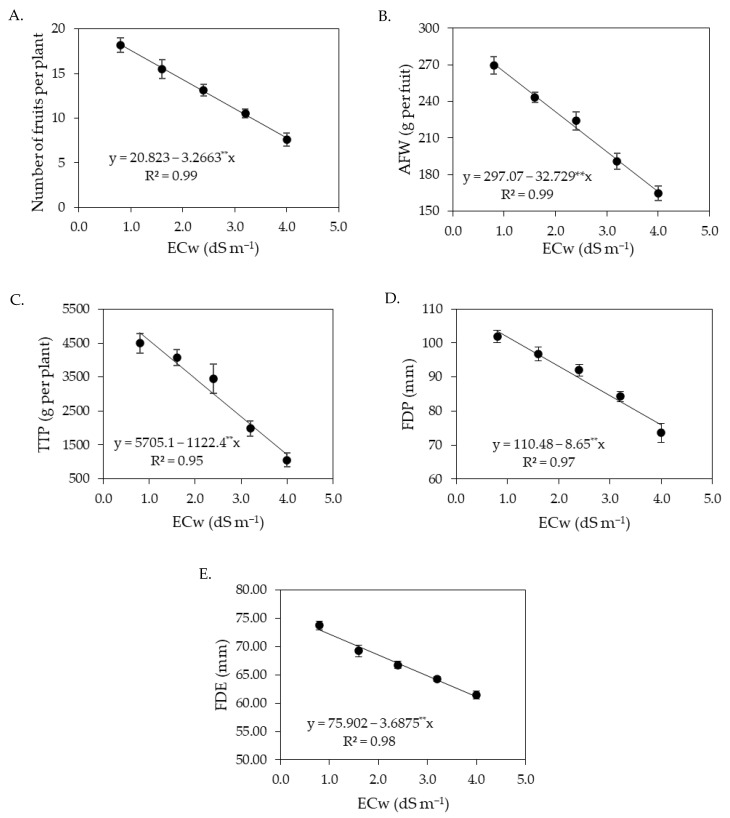
Number of fruits per plant—NFP (**A**), average fruit weight—AFW (**B**), total production per plant—TPP (**C**), fruit polar diameter—FPD (**D**), and fruit equatorial diameter—FED (**E**) of sour passion fruit as a function of the levels ECw in the second production cycle. ** Significant at *p* ≤ 0.01. Vertical lines represent standard error of the mean (*n* = 3).

**Figure 14 plants-12-02023-f014:**
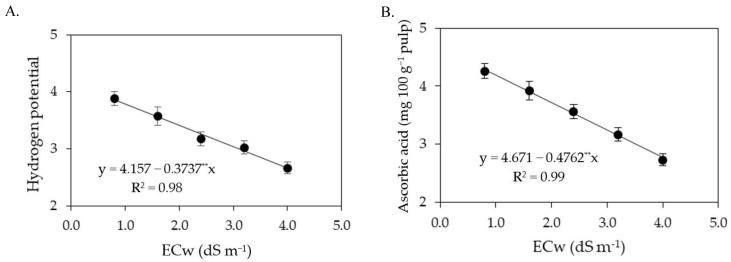
Hydrogen potential—pH (**A**), ascorbic acid—AA (**B**), soluble solids—SS (**C**), and titratable acidity—TA (**D**) in the pulp of sour passion fruit as a function of the levels ECw in the second production cycle. ** Significant at *p* ≤ 0.01. Vertical lines represent standard error of the mean (*n* = 3).

**Figure 15 plants-12-02023-f015:**
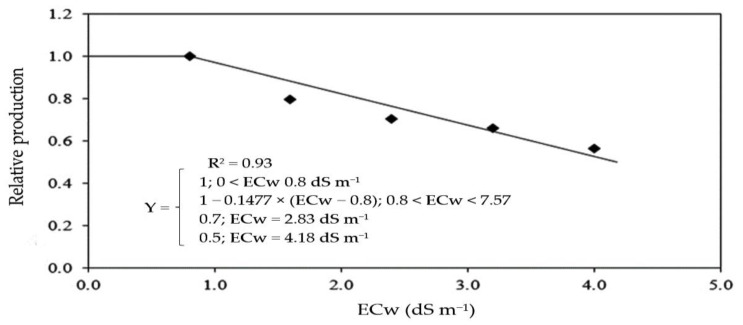
Relative production of sour passion fruit as a function of the irrigation water salinity (ECw), described by the plateau followed by linear decline model proposed by Maas and Hoffman [74], calculated considering the production values obtained at an ECw from 0.8 to 4.0 dS m^−1^.

**Figure 16 plants-12-02023-f016:**
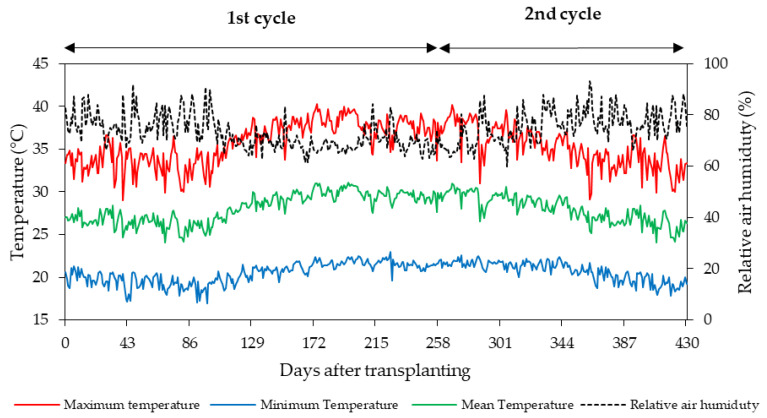
Meteorological data collected during the experimental period.

**Figure 17 plants-12-02023-f017:**
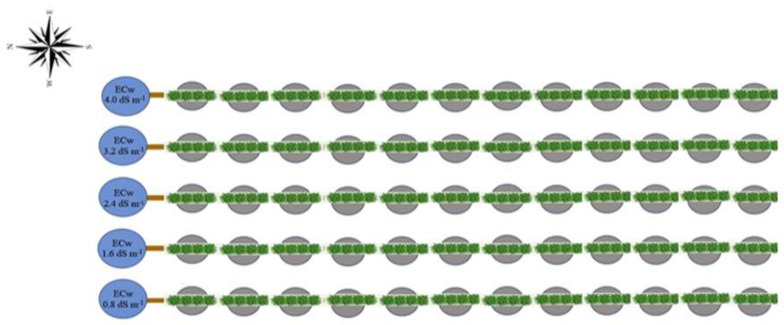
Layout of the experimental area.

**Figure 18 plants-12-02023-f018:**
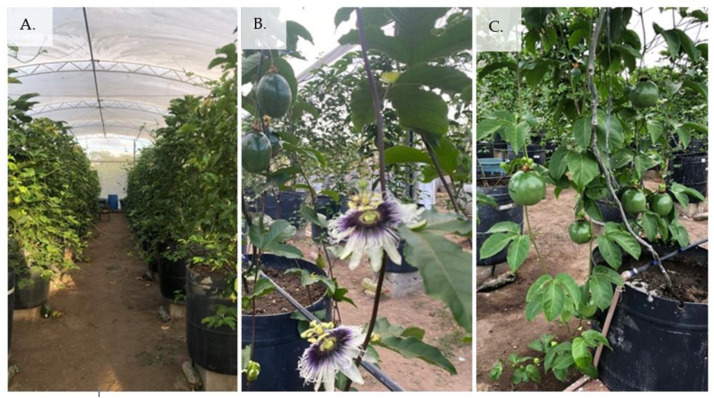
Sour passion fruits in different stages of development (Vegetative stage—(**A**), flowering stage—(**B**), and fruiting stage—(**C**)).

**Table 1 plants-12-02023-t001:** Summary of the analysis of variance for the relative water content and electrolyte leakage in the leaf blade of sour passion fruit, under different levels of electrical conductivity of irrigation water and salicylic acid, at 180 days after transplanting (DAT).

SV	DF	Mean Squares	
		RWC	% EL
Electrical conductivity of water (ECw)	4	831.35 **	48.73 **
Linear regression	1	2320.93 **	120.89 **
Quadratic regression	1	119.24 ^ns^	24.98 ^ns^
Residual 1	8	15.68	0.47
Salicylic acid (SA)	3	107.04 **	6.58 *
Linear regression	1	167.45 *	0.14 ^ns^
Quadratic regression	1	130.47 **	19.48 **
Interaction (ECw × SA)	12	75.94 **	8.36 **
Residual 2	32	11.38	2.79
CV 1 (%)		5.29	5.69
CV 2 (%)		4.51	13.87

SV—Source of variation; DF—degree of freedom; CV (%)—coefficient of variation; * significant at 0.05 level of probability; ** significant at 0.01 level of probability; and ns—not significant.

**Table 2 plants-12-02023-t002:** Summary of the analysis of variance for internal CO_2_ concentration, stomatal conductance, transpiration, and CO_2_ assimilation rate of sour passion fruit under different levels of electrical conductivity of irrigation water and salicylic acid, at 180 DAT.

SV	DF	Mean Squares			
		*Ci*	*gs*	*E*	*A*
Electrical conductivity of water (ECw)	4	56,260.48 **	1.74 × 10^−2^ **	1.07 **	5.84 **
Linear regression	1	211,848.03 **	3.97 × 10^−2^ **	2.29 **	9.09 *
Quadratic regression	1	11,666.67 ^ns^	9.32 × 10^−3^ **	1.16 *	10.14 **
Residual 1	8	26.72	1.40 × 10^−5^	4.63 × 10^−4^	6.11 × 10^−3^
Salicylic acid (SA)	3	2916.38 **	1.03 × 10^−3^ **	0.23 **	1.04 **
Linear regression	1	5843.25 *	5.47 × 10^−4^ **	0.12 ^ns^	0.30 ^ns^
Quadratic regression	1	1926.67 **	1.93 × 10^−3^ **	0.33 **	1.25 **
Interaction (ECw × SA)	12	181.79 **	6.20 × 10^−5^ **	5.49 × 10^−3^ **	0.03 **
Residual 2	32	67.74	1.62 × 10^−4^	1.33 × 10^−2^	0.54
CV 1 (%)		2.51	2.78	1.78	3.57
CV 2 (%)		4.00	6.01	5.62	14.72

SV—Source of variation; DF—degree of freedom; CV (%)—coefficient of variation; * significant at 0.05 level of probability; ** significant at 0.01 level of probability; and ns—not significant.

**Table 3 plants-12-02023-t003:** Summary of the analysis of variance for Chl *a*—chlorophyll *a*, Chl *b*—chlorophyll *b*, Chl *t*—total chlorophyll, and Car—carotenoids of sour passion fruit under different levels of electrical conductivity of irrigation water and salicylic acid at 180 DAT.

SV	DF	Mean Squares			
		Chl *a*	Chl *b*	Chl *t*	Car
Electrical conductivity of water (ECw)	4	90,590.46 **	12,497.44 **	169,455.31 **	12,562.68 **
Linear regression	1	227,041.75 **	38,137.66 **	451,219.54 **	48,547.99 **
Quadratic regression	1	102,661.66 ^ns^	8207.09 ^ns^	168,925.51 ^ns^	1062.09 ^ns^
Residual 1	8	532.88	11.78	510.73	0.07
Salicylic acid (SA)	3	85,227.58 **	8162.89 **	144,564.62 **	89.76 ^ns^
Linear regression	1	90,129.43 ^ns^	4531.15 ^ns^	149,529.27 *	-
Quadratic regression	1	102,004.18 **	12,952.76 **	171,415.21 **	-
Interaction (ECw × SA)	12	1649.41 *	54.15 *	1915.92 *	0.53 ^ns^
Residual 2	32	1482.89	124.69	1501.21	0.69
CV 1 (%)		3.71	3.89	2.81	6.14
CV 2 (%)		6.19	6.72	4.82	8.44

SV—Source of variation; DF—degree of freedom; CV (%)—coefficient of variation; * significant at 0.05 of probability; ** significant at 0.01 level of probability; and ns—not significant.

**Table 4 plants-12-02023-t004:** Summary of the analysis of variance for F_0_—initial fluorescence Fm—maximum fluorescence, Fv—variable fluorescence, and Fv/Fm—quantum efficiency of photosystem II of sour passion fruit irrigated with saline water and salicylic acid at 180 DAT.

SV	DF	Mean Squares			
		F_0_	Fm	Fv	Fv/Fm
Electrical conductivity of water (ECw)	4	3981.23 **	120,560.85 **	182,939.77 **	0.006 **
Linear regression	1	1587.00 **	47,270.21 **	72,199.53 **	0.023 **
Quadratic regression	1	54.85 ^ns^	2941.72 ^ns^	2468.67 ^ns^	0.001 ^ns^
Residual 1	8	91.65	20,781.50	23,766.24	0.0004
Salicylic acid (SA)	3	78.86 ^ns^	6650.60 ^ns^	6837.17 ^ns^	0.0002 ^ns^
Interaction (ECw × SA)	12	57.33 ^ns^	3763.25 ^ns^	4501.81 ^ns^	0.0001 ^ns^
Residual 2	32	209.23	41,204.07	45,247.04	0.0007
CV 1 (%)		1.43	6.61	9.48	2.72
CV 2 (%)		2.16	9.31	13.08	3.67

SV—Source of variation; DF—degree of freedom; CV (%)—coefficient of variation; ** significant at 0.01 level of probability; and ns—not significant.

**Table 5 plants-12-02023-t005:** Summary of the analysis of variance for the NFP—number of fruits, TPP—total production per plant, AFW—average fruit weight, FPD—fruit polar diameter, and FED—fruit equatorial diameter of sour passion fruit irrigated with saline water and salicylic acid.

SV	DF	Mean Squares				
		NFP	TPP	AFW	FPD	FED
Electrical conductivity of water (ECw)	4	435.32 **	45,478,334.44 **	23,256.97 **	1192.04 **	133.01 **
Linear regression	1	1736.75 **	177,682,003.33 **	76,081.37 **	4371.51 **	480.84 **
Quadratic regression	1	2.24 ^ns^	4,163,582.69 ^ns^	15,074.53 ^ns^	240.24 ^ns^	0.04 ^ns^
Residual 1	8	0.50	257,253.92	169.15	0.09	17.43
Salicylic acid (SA)	3	26.81 **	1,468,987.39 *	1431.03 **	185.75 ^ns^	26.06 ^ns^
Linear regression	1	0.57 ^ns^	666,382.38 ^ns^	1106.22 ^ns^	-	-
Quadratic regression	1	40.13 **	2,088,240.70 **	3011.83 **	-	-
Interaction (ECw × SA)	12	0.89 ^ns^	101,170.49 ^ns^	252.14 ^ns^	24.88 ^ns^	3.40 ^ns^
Residual 2	32	2.67	433,069.83	626.52	31.78	49.05
CV 1 (%)		3.83	11.59	5.69	3.36	6.37
CV 2 (%)		8.85	15.03	10.96	6.73	10.69

SV—Source of variation; DF—degree of freedom; CV (%)—coefficient of variation; * significant at 0.05 level of probability; ** significant at 0.01 level of probability; and ns—not significant.

**Table 6 plants-12-02023-t006:** Summary of the analysis of variance for the pH—hydrogen potential, SS—soluble solids, AA—ascorbic acid, and TA—titratable acidity of pulp of sour passion fruit irrigated with saline water and salicylic acid in the first production cycle.

SV	DF	Mean Squares			
		pH	SS	AA	TA
Electrical conductivity of water (ECw)	4	1.10 × 10^−2^ **	12.91 **	0.73 **	6.83 **
Linear regression	1	-	48.83 **	0.11 *	25.60 **
Quadratic regression	1	-	2.55 ^ns^	0.04 ^ns^	1.63 *
Residual 1	8	4.43 × 10^−3^	0.40	0.24	0.21
Salicylic acid (SA)	3	9.93 × 10^−2^ **	5.90 **	3.64 **	3.49 **
Linear regression	1	0.08 *	4.13 *	2.41 ^ns^	2.38 *
Quadratic regression	1	0.12 **	10.88 **	7.28 **	6.36 **
Interaction (ECw × SA)	12	3.10 × 10^−2^ **	0.04 ^ns^	2.02 **	0.02 ^ns^
Residual 2	32	2.47 × 10^−3^	0.93	0.19	0.51
CV 1 (%)		2.73	4.44	16.76	4.37
CV 2 (%)		2.04	6.77	15.08	6.72

SV—Source of variation; DF—degree of freedom; CV (%)—coefficient of variation; * significant at 0.05 level of probability; ** significant at 0.01 level of probability; and ns—not significant.

**Table 7 plants-12-02023-t007:** Summary of the analysis of variance for the RWC—relative water content and EL—electrolyte leakage in the leaf blade of sour passion fruit under different levels of electrical conductivity of irrigation water and salicylic acid at 360 DAT.

SV	DF	Mean Squares	
		RWC	% EL
Electrical conductivity of water (ECw)	4	831.35 **	48.73 **
Linear regression	1	2320.93 **	120.89 **
Quadratic regression	1	119.24 ^ns^	24.98 ^ns^
Residual 1	8	15.68	0.47
Salicylic acid (SA)	3	107.04 **	6.58 *
Interaction (ECw × SA)	12	75.94 **	8.36 **
Residual 2	32	11.38	2.79
CV 1 (%)		5.29	5.69
CV 2 (%)		4.51	13.87

SV—Source of variation; DF—degree of freedom; CV (%)—coefficient of variation; * significant at 0.05 level of probability; ** significant at 0.01 level of probability; and ns—not significant.

**Table 8 plants-12-02023-t008:** Summary of the analysis of variance for the internal CO_2_ concentration, stomatal conductance, transpiration, and CO_2_ assimilation rate of sour passion fruit under different levels of electrical conductivity of irrigation water and salicylic acid at 360 DAT.

SV	DF	Mean Squares			
		*Ci*	*gs*	*E*	*A*
Electrical conductivity of water (ECw)	4	47,673.22 **	1.01 × 10^−2^ **	0.76 **	3.36 **
Linear regression	1	178,926.46 **	3.91 × 10^−2^ **	2.92 **	12.56 **
Quadratic regression	1	10,232.04 *	1.62 × 10^−3 ns^	0.05 ^ns^	0.03 ^ns^
Residual 1	8	118.81	1.20 × 10^−5^	0.03	0.08
Salicylic acid (SA)	3	1374.21 ^ns^	5.25 × 10^−4 ns^	0.02 ^ns^	0.30 ^ns^
Interaction (ECw × SA)	12	170.87 ^ns^	6.10 × 10^−5 ns^	0.03 ^ns^	0.13 ^ns^
Residual 2	32	71.62	1.42 × 10^−4^	0.04	0.19
CV 1 (%)		5.87	3.02	3.11	6.90
CV 2 (%)		4.56	7.12	3.40	10.91

SV—Source of variation; DF—degree of freedom; CV (%)—coefficient of variation; * significant at 0.05 level of probability; ** significant at 0.01 level of probability; and ns—not significant.

**Table 9 plants-12-02023-t009:** Summary of the analysis of variance for chlorophyll *a* (Chl *a*), chlorophyll *b* (Chl *b*), total chlorophyll (Chl *t*), and carotenoids (Car) of sour passion fruit under different levels of ECw and SA at 180 DAT.

SV	DF	Mean Squares			
		Chl *a*	Chl *b*	Chl *t*	Car
Electrical conductivity of water (ECw)	4	42,083.82 **	5349.26 **	77,411.91 **	9076.55 **
Linear regression	1	162,243.95 **	20,613.90 **	29,858.83 **	3507.96 **
Quadratic regression	1	1791.71 ^ns^	378.81 ^ns^	3818.19 *	767.36 ^ns^
Residual 1	8	409.62	162.64	613.14	0.38
Salicylic acid (SA)	3	361.27 ^ns^	463.24 ^ns^	1567.08 ^ns^	64.85 ^ns^
Interaction (ECw × SA)	12	419.26 ^ns^	124.62 ^ns^	587.54 ^ns^	0.86 ^ns^
Residual 2	32	263.43	36.46	261.37	0.42
CV 1 (%)		4.68	10.76	4.45	6.39
CV 2 (%)		3.64	8.24	2.91	5.41

SV—Source of variation; DF—degree of freedom; CV (%)—coefficient of variation; * significant at 0.05 level of probability; ** significant at 0.01 of probability; and ns—not significant.

**Table 10 plants-12-02023-t010:** Summary of the analysis of variance for the F_0_—initial fluorescence, Fm—maximum fluorescence, Fv—variable fluorescence, and Fv/Fm— quantum efficiency of photosystem II of sour passion fruit irrigated with saline water and salicylic acid at 360 DAT.

SV	DF	Mean Squares			
		F_0_	Fm	Fv	Fv/Fm
Electrical conductivity of water (ECw)	4	3686.68 **	60,774.36 **	119,387.17 **	0.017 **
Linear regression	1	18,682.04 **	23,829.83 **	472,408.49 ^ns^	0.068 **
Quadratic regression	1	64.55 ^ns^	1482.27 ^ns^	1123.57 ^ns^	0.0001 ^ns^
Residual 1	8	107.87	10,476.11	13,380.23	0.001
Salicylic acid (SA)	3	92.77 ^ns^	3352.38 ^ns^	3628.06 ^ns^	0.0004 ^ns^
Interaction (ECw × SA)	12	67.49 ^ns^	1897.02 ^ns^	2620.32 ^ns^	0.0003 ^ns^
Residual 2	32	246.29	20,770.58	24,883.95	0.002
CV 1 (%)		2.63	8.20	12.78	3.97
CV 2 (%)		3.25	11.55	17.43	5.75

SV—Source of variation; DF—degree of freedom; CV (%)—coefficient of variation; ** significant at 0.01 level of probability; and ns—not significant.

**Table 11 plants-12-02023-t011:** Summary of the analysis of variance for the NFP—number of fruits per plant, TPP—total production per plant, AFW—average fruit weight (AFW), FPD—fruit polar diameter, and FED—fruit equatorial diameter of sour passion fruit irrigated with saline water and salicylic acid.

SV	DF	Mean Squares				
		NFP	TPP	PMF	FPD	FED
Electrical conductivity of water (ECw)	4	213.27 **	2,589,259.98 **	42,055.93 **	1390.49 **	146.61 **
Linear regression	1	850.88 **	9,675,233.51 ^ns^	82,206.77 **	5099.16 **	530.04 **
Quadratic regression	1	1.10 ^ns^	4,341,362.39 ^ns^	74,558.72 ^ns^	280.24 ^ns^	0.04 ^ns^
Residual 1	8	0.24	55,776.44	295.74	0.11	19.21
Salicylic acid (SA)	3	13.12 ^ns^	109,854.77 ^ns^	2214.24 ^ns^	216.79	28.76 ^ns^
Interaction (ECw × SA)	12	0.43 ^ns^	328,197.07 ^ns^	2890.53 ^ns^	29.03 ^ns^	2.75 ^ns^
Residual 2	32	1.31	238,300.49	1210.20	37.06	54.08
CV 1 (%)		3.83	7.74	7.92	2.76	5.77
CV 2 (%)		8.85	16.00	15.02	5.73	9.69

SV—Source of variation; DF—degree of freedom; CV (%)—coefficient of variation; ** significant at 0.01 level of probability; and ns—not significant.

**Table 12 plants-12-02023-t012:** Summary of the analysis of variance for the pH—hydrogen potential, SS—soluble solids, AA—ascorbic acid, and TA—titratable acidity in the pulp of sour passion fruit irrigated with saline water and salicylic acid in the second production cycle.

SV	DF	Mean Squares			
		pH	SS	AA	TA
Electrical conductivity of water (ECw)	4	2.74 **	14.15 **	4.34 **	16.56 **
Linear regression	1	10.82 **	55.06 **	17.33 **	63.90 **
Quadratic regression	1	0.01 ^ns^	1.24 ^ns^	0.05 ^ns^	2.31 ^ns^
Residual 1	8	1.17 × 10^−4^	0.19	0.002	0.001
Salicylic acid (SA)	3	0.15 ^ns^	0.44 ^ns^	0.06 ^ns^	0.05 ^ns^
Interaction (ECw × SA)	12	6.64 × 10^−4 ns^	0.13 ^ns^	0.03 ^ns^	0.01 ^ns^
Residual 2	32	2.47 × 10^−3^	0.27	0.005	0.02
CV 1 (%)		2.40	3.09	3.40	3.18
CV 2 (%)		3.88	3.70	5.02	4.88

SV—Source of variation; DF—degree of freedom; CV (%)—coefficient of variation; ** significant at 0.01 level of probability; and ns—not significant.

## Data Availability

Not applicable.

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
