# Peer review of "Foliar Applications of Salicylic Acid on Boosting Salt Stress Tolerance in Sour Passion Fruit in Two Cropping Cycles"

_plants, 2023, doi:10.3390/plants12102023_

Round 1
Reviewer 1 Report
Dear Authors
Despite the fact that it is an excellent topic and research, it is not well presented.
In the abstract, please provide more information about the results. There are only five lines describing the methods.
It would be helpful to include crop names in the introduction, especially when the finding is related to passion fruits, like line 46 (reference 9).
There is no coherence in the introduction. Provide a coherent and related structure to the paragraphs.
I completely agree that charts are more professional, but I find them a bit difficult to understand.
Regards
Author Response
Campina Grande, PB
May, 15, 2023
Reference: Plants - 2401401 - Response to Review Report 1
Dear Editor
The authors are very grateful to you and the Reviewers for the positive and constructive comments and suggestions on our manuscript entitled “Salicylic Acid on the Induction of Salt Stress Tolerance in Sour Passion Fruit in Two Cropping Cycles”. The authors would like to inform you that a thorough revision of the manuscript was made, incorporating the suggestions and adapting the text according to the comments. Attached is the revised version of the manuscript. All changes in the text are highlighted in red color.
The authors remain at your disposal for any further information and explanation.
The responses/clarifications to the issues raised by the Reviewer 1/Editor are presented below:
REVIEWER 1
- In the abstract, please provide more information about the results. There are only five lines describing the methods.
Response: According to the normas of ‘Plants, the abstract must have a maximum of 200 words, providing more information on results will exceed the 200 word limit. For this reason, it was not possible to comply with the reviewer's suggestion.
- It would be helpful to include crop names in the introduction, especially when the finding is related to passion fruits, like line 46 (reference 9)
Response: The suggestion was incorporated in the revised version of the manuscript, as can be seen between lines 49 and 54.
- There is no coherence in the introduction. Provide a coherent and related structure to the paragraphs.
Response: Dear reviewer, authors prepared the introduction following a logical sequence of arguments: initially contextualizing the research problem and its consequence in the cultivation of passion fruit. Then, the ways of mitigating salinity through the application of salicylic acid were reinforced through the state of the art, demonstrating examples with several fruit crops. Soon after, the importance of passion fruit cultivation was justified and the research hypothesis was presented as a state-of-the-art premise. Finally, a solution is proposed through the objective. However, if necessary, we can modify the sequence by bringing the question of importance of passion fruit crop in the beginning.
- I completely agree that charts are more professional, but I find them a bit difficult to understand.
Response: The response surface graphs are recommended to explain the effect of the interaction between two quantitative factors, allowing the visualization of the maximum and minimum points on the surface, in addition, through the regression equation it is possible to estimate possible response for any given value within the study interval.
Yours sincerely,
Geovani Soares de Lima
Corresponding author

Reviewer 2 Report
The article presents many data on the important academic value of the plants, both at the physiological level and at the production and agronomic level. Despite the fact that they study the effect of salicylic acid on salinity and its relationship and anti-saline stress effect is already known, they give it a good focus by studying the two production batches and using a lesser-known fruit such as the sour passion fruit. I don't have many comments or objections, but I'll leave you with some ideas below. In general, it is a good article, rigorous in the method, and where the main parameters have been looked at. However, I have missed a characterization of the levels of ac. salicylic in the plant, to see the actual amount that is being applied. On the other hand, the work would have been rounder if some of the genes associated with these stresses had been looked at and tried to correlate the issue of water, alteration of water transport due to high salinity, genes that produce free sugars that act as osmolytes in the fruit. 5 to 10 RT-qPCRs and for me this article was perfect. Congratulations on your work.
Author Response
Campina Grande, PB
May, 15, 2023
Reference: Plants - 2401401 - Response to Review Report 2
Dear Editor
The authors are very grateful to you and the Reviewers for the positive and constructive comments and suggestions on our manuscript entitled “Salicylic Acid on the Induction of Salt Stress Tolerance in Sour Passion Fruit in Two Cropping Cycles”. The authors would like to inform you that a thorough revision of the manuscript was made, incorporating the suggestions and adapting the text according to the comments. Attached is the revised version of the manuscript. All changes in the text are highlighted in red color.
The authors remain at your disposal for any further information and explanation.
The responses/clarifications to the issues raised by the Reviewer 2/Editor are presented below:
REVIEWER 2
- The article presents many data on the important academic value of the plants, both at the physiological level and at the production and agronomic level. Despite the fact that they study the effect of salicylic acid on salinity and its relationship and anti-saline stress effect is already known, they give it a good focus by studying the two production batches and using a lesser-known fruit such as the sour passion fruit. I don't have many comments or objections, but I'll leave you with some ideas below. In general, it is a good article, rigorous in the method, and where the main parameters have been looked at. However, I have missed a characterization of the levels of ac. salicylic in the plant, to see the actual amount that is being applied. On the other hand, the work would have been rounder if some of the genes associated with these stresses had been looked at and tried to correlate the issue of water, alteration of water transport due to high salinity, genes that produce free sugars that act as osmolytes in the fruit. 5 to 10 RT-qPCRs and for me this article was perfect. Congratulations on your work
Response: Authors thank the reviewer for the compliments and suggestions regarding our manuscript and remain at disposal to clarify any doubts, in future studies we can incorporate in objectives the suggested points.
Yours sincerely,
Geovani Soares de Lima
Corresponding author

Reviewer 3 Report
The study is quite interesting. The results have significant applications in the fruit industry. Please refer to the enclosed Word file for suggested edits.

Author Response
Campina Grande, PB
May, 15, 2023
Reference: Plants - 2401401 - Response to Review Report 3
Dear Editor
The authors are very grateful to you and the Reviewers for the positive and constructive comments and suggestions on our manuscript entitled “Salicylic Acid on the Induction of Salt Stress Tolerance in Sour Passion Fruit in Two Cropping Cycles”. The authors would like to inform you that a thorough revision of the manuscript was made, incorporating the suggestions and adopting the text according to the comments. Attached is the revised version of the manuscript. All changes in the text are highlighted in red color.
The authors remain at your disposal for any further information and explanation.
The responses/clarifications to the issues raised by the Reviewer 3/Editor are presented below:
REVIEWER 3
- The study is quite interesting. The results have significant applications in the fruit industry. Please refer to the enclosed Word file for suggested edits.
Response: Authors thank the reviewer for the compliments regarding our manuscript and remain at disposal to clarify any doubts.
- Which ones? Please list some examples.
Response: The introduction excerpt was reformulated in the revised version of the manuscript, inserting examples as suggested by the reviewer, as can be seen between lines 49 and 54.
- - Vitis labrusca or grapevines in America?
Response: The author, in the manuscript, used the term American grapevine.
Ekbic, H. B.; Ozcan, N.; Erdem, H. Impacts of salicylic acid treatments on salt resistance of some American grapevine rootstocks. Fresenius Environ. Bull. 2020, 29, 685-692.
Yours sincerely,
Geovani Soares de Lima
Corresponding author

Reviewer 4 Report
The manuscript by Sobrinho et al. investigated the salicylic acid on the induction of salt stress tolerance in sour passion fruit in two cropping cycles, it is an interesting topic, but there are some drawbacks in the manuscript.
1, In the Abstract part, the authors should briefly introduce the research background and the significance of the study.
2, P8, in Figure 4, the ECw is not a continuous variable in this study, therefore, it's better to present the data in a scatter plot form or column chart.
I am curious that why the linear equation can be used to fit the relationship between ECw and chlorophyll fluorescence parameters, what is the principle? Maybe the linear relationship is just observed in this study, or it is just an accidental phenomenon.
I also noticed that the linear or non-linear equations were used to fit the relationships between the ECw or salicylic acid concentrations and other parameters in many figures. The authors need to explain why the equations can be used to fit those relationships.
3, In Figure 10, the unit of Ci should be μmol mol-1.
4, In the Discussion part, the authors should also discuss the interactions between the RWC, EL, photosynthesis, chlorophyll fluorescence parameters, plant growth and fruit quality under salt stress and foliar application of salicylic acid, but not just discuss those indices separately.
And this section should better avoid the repetition of the results, but focus on the discussion of the influence and attenuate mechanisms of salt stress deleterious on the sour passion fruit.
Furthermore, if the authors could give suggestions on the practical application of salicylic acid in the field cultivation when the crops or fruit trees suffer from salt stress, that would be better.
Author Response
Campina Grande, PB
May, 15, 2023
Reference: Plants - 2401401 - Response to Review Report 4
Dear Editor
The authors are very grateful to you and the Reviewers for the positive and constructive comments and suggestions on our manuscript entitled “Salicylic Acid on the Induction of Salt Stress Tolerance in Sour Passion Fruit in Two Cropping Cycles”. The authors would like to inform you that a thorough revision of the manuscript was made, incorporating the suggestions and adapting the text according to the comments. Attached is the revised version of the manuscript. All changes in the text are highlighted in red color.
The authors remain at your disposal for any further information and explanation.
The responses/clarifications to the issues raised by the Reviewer 4/Editor are presented below:
REVIEWER 4
- In the Abstract part, the authors should briefly introduce the research background and the significance of the study
Response: According to the norms of ‘Plants’, the abstract must have a maximum of 200 words, providing more information to the abstract will exceed the limit of 200 words, for this reason, it was not possible to comply with the reviewer's suggestion.
- P8, in Figure 4, the ECw is not a continuous variable in this study, therefore, it's better to present the data in a scatter plot form or column chart.
Response: Dear reviewer, Figure 4 deals with the analysis of quantitative factors (levels of electrical conductivity of irrigation water and concentrations of salicylic acid), thus, the use of regression analysis becomes more appropriate to present the results, as it permits to visualize possible minimum and maximum response of the factors in question. The use of a column graphs, in our opinion, would be more appropiate to analyze qualitative factors, for this reason the graphs were not reformulated.
- I am curious that why the linear equation can be used to fit the relationship between ECw and chlorophyll fluorescence parameters, what is the principle? Maybe the linear relationship is just observed in this study, or it is just an accidental phenomenon.
Response: As described in item 4.6 of the revised manuscript, the choice of model (linear or quadratic) was based on the coefficients of determination (R2).
- I also noticed that the linear or non-linear equations were used to fit the relationships between the ECw or salicylic acid concentrations and other parameters in many figures. The authors need to explain why the equations can be used to fit those relationships
Response: Regression analysis is one of the most used techniques and allows modeling a relationship between two sets of variables, the equation generated in the analysis can be used to make projections or data estimates. Furthermore, regression analysis allows us to determine the degree to which the independent variables influence the dependent variables. The choice of model was based on the coefficient of determination (R2). Similar effects were also observed by Capitulino et al. (2023).
https://doi.org/10.3390/plants12030599
- Na Figura 10, a unidade de Ci deve ser μmol mol -1
Response: Thanks, the unit of Ci has been reformulated in the revised version of the manuscript.
- In the Discussion section, authors should also discuss the interactions between RWC, EL, photosynthesis, chlorophyll fluorescence parameters, plant growth and fruit quality under salt stress and foliar application of salicylic acid, but not just discuss these indices separately
Response: The suggestion was met in the revised version of the manuscript, as can be seen between the lines 687 e 696.
- And this section should better avoid the repetition of the results, but focus on the discussion of the influence and attenuate mechanisms of salt stress deleterious on the sour passion fruit.
Response: The discussion item addressed the explanation of the results obtained in the research based on the hypothesis under study, as well as its comparison with other findings in the literature, always seeking to avoid the repetition of results.
- Furthermore, if the authors could give suggestions on the practical application of salicylic acid in the field cultivation when the crops or fruit trees suffer from salt stress, that would be better.
Response: Dear reviewer, the results obtained in this study refer to research in a protected environment (Greenhouse). There is no information in the literature about the practical application of salicylic acid in cultivation under field conditions.
Yours sincerely,
Geovani Soares de Lima
Corresponding author
